# FASTER HYPERPARAMETER SEARCH FOR GNNs VIA CALIBRATED DATASET CONDENSATION

## ABSTRACT

Dataset condensation aims to reduce the computational cost of training multiple models on a large dataset by condensing the training dataset into a small synthetic set. State-of-the-art approaches rely on matching the model gradients for the real and synthetic data and have recently been applied to condense large-scale graphs for node classification tasks. Although dataset condensation may be efficient when training multiple models for hyperparameter optimization, there is no theoretical guarantee on the generalizability of the condensed data: data condensation often generalizes poorly *across hyperparameters/architectures* in practice, while we find and prove this overfitting is much more severe on graphs. In this paper, we consider a different condensation objective specifically geared towards *hyperparameter search*. We aim to generate the synthetic dataset so that the validation-performance rankings of the models, with different hyperparameters, on the condensed and original datasets are comparable. We propose a novel *hyperparameter-calibrated dataset condensation* (HCDC) algorithm, which obtains the synthetic validation data by matching the *hyperparameter gradients* computed via implicit differentiation and efficient inverse Hessian approximation. HCDC employs a supernet with differentiable hyperparameters, making it suitable for modeling GNNs with widely different convolution filters. Experiments demonstrate that the proposed framework effectively maintains the validation-performance rankings of GNNs and speeds up hyperparameter/architecture search on graphs.

## 1 INTRODUCTION

Graph neural networks (GNNs) have found remarkable success in tackling a variety of graph-related tasks (Hamilton, 2020). However, the prevalence of large-scale graphs in real-world contexts, such as social, information, and biological networks (Hu et al., 2020), which frequently scale up to millions/billions of nodes and edges, poses significant computational issues for training GNNs. While training a single model can be expensive, designing deep learning models for new tasks requires substantially more computation, as this involves training multiple models on the same dataset many times to verify the design choice (e.g., the architecture and hyperparameter choice (Elsken et al., 2019)). Towards this end, we consider the following question: how can we reduce the computational cost for training multiple models on the same dataset, for *hyperparameter search/optimization*?

Natural approaches to reduce the training set size include methods such as graph coreset selection (Baker et al., 2020), graph sparsification (Batson et al., 2013), graph coarsening (Loukas, 2019) and graph sampling (Zeng et al., 2019). However, all of these methods involve selecting samples from the given training set, which limits the performance. A more effective alternative is to *synthesize* informative samples rather than select from the given samples. *Dataset condensation* (Zhao et al., 2020) has emerged as a competent data mechanism to synthesize data, with promising results. It aims to produce a small synthetic training set such that a model trained on the synthetic set obtains testing accuracy comparable to that trained on the original training set.

Although dataset condensation achieves the state-of-the-art performance for neural networks trained on condensed samples, this technique is inadequate for accelerating hyperparameter search/optimization, as: (1) theoretically, dataset condensation obtains synthetic samples that minimize the performance drop of a specific model; however, there is no performance guarantee when using this condensed data to train other models, and (2) in practice, it is unclear how condensation

methods compare with strong baselines such as various coreset methods, in terms of their ability to preserve the outcome of architecture/hyperparameter optimization.

In this paper, we identify the poor generalizability of existing condensed data approaches on *graphs* (Jin et al., 2021) *across architectures/hyperparameters*, as this topic has been overlooked in the existing literature, which focuses more on image condensation. We prove that graph condensation fails to preserve validation performance rankings of GNN architectures, and identify two dominant reasons for this failure: (1) most GNNs differ from each other in terms of their convolution filter design. Thus, when performing condensation with a single GNN, the condensed data is overfitted to the corresponding GNN filter, a single biased point in the set of GNN filters; and (2) the learned adjacency matrix of the synthetic graph considerably overfits the condensed data, and thus fails to maintain the characteristics of the original adjacency matrix.

To solve the poor generalizability issue, we develop a new dataset condensation framework that *preserves the outcome of hyperparameter search/optimization* on the condensed data. We propose to learn synthetic data as well as its validation split such that the validation performance ranking of architectures on the synthetic and original datasets are comparable.

Under the assumption of a continuous hyperparameter space or a generic supernet which interpolates all architectures, we find and prove that the goal of preserving validation performance rankings can be realized by matching the *hyperparameter gradients* on the synthetic and original validation data. The hyperparameter gradients (or *hypergradients* for short) can be efficiently computed with constant memory overhead by the implicit function theorem (IFT) and the Neumann series approximation of an inverse Hessian (Lorraine et al., 2020). Consequently, we propose a *hyperparameter calibrated dataset condensation* (HCDC) framework assuming continuous hyperparameters, which is suitable to modeling GNNs with different convolution matrices. Experiments demonstrate the effectiveness of the proposed framework in preserving the performance rankings of GNNs.

Although beyond the scope of this paper, HCDC also has the potential to be combined with the supernets in differentiable neural architecture search (differentiable NAS) methods (Liu et al., 2018) to tackle the general neural architecture space for image and text data.

Our contributions can be summarized as follows: **(1)** We formulate a new dataset condensation objective for hyperparameter optimization and propose the *hyperparameter calibrated dataset condensation* (HCDC) framework that learns synthetic validation data by matching the hypergradients. **(2)** We prove the hardness of generalizing the condensed graph across GNN architectures, and the validity of HCDC in preserving the validation performance rankings of GNNs. **(3)** Experiments demonstrate the effectiveness of HCDC in further reducing the search time of off-the-shelf graph NAS algorithms, from several hours to minutes on graphs with millions of nodes.

## 2 SETTINGS, BACKGROUND, AND CHALLENGES

### 2.1 SETTINGS: NODE CLASSIFICATION AND GNNS

This paper adopts graph learning notations, but HCDC is generally applicable to other data, tasks, and models; see Appendix B for discussions.

**Node classification on a graph** considers a graph $\mathcal{T} = (A, X, \mathbf{y})$ with adjacency matrix $A \in \{0, 1\}^{n \times n}$, node features $X \in \mathbb{R}^{n \times d}$, node class labels $\mathbf{y}$, and mutually disjoint node-splits $V_{\text{train}} \bigcup V_{\text{val}} \bigcup V_{\text{test}} = [n]$. Using a *graph neural network* (GNN) $f_{\theta,\lambda} : \mathbb{R}_{\geq 0}^{n \times n} \times \mathbb{R}^{n \times d} \to \mathbb{R}^{n \times K}$, where $\theta \in \Theta$ denotes the parameters and $\lambda \in \Lambda$ denotes the hyper-parameters (if they exist), we aim to find $\theta^{\mathcal{T}} = \arg\min_{\theta} \mathcal{L}_{\mathcal{T}}^{\text{train}}(\theta, \lambda)$, where $\mathcal{L}_{\mathcal{T}}^{\text{train}}(\theta, \lambda) := \sum_{i \in V_{\text{train}}} \ell([f_{\theta,\lambda}(A, X)]_i, y_i)$ and $\ell(\hat{y}, y)$ is the cross-entropy loss. The node classification loss $\mathcal{L}_{\mathcal{T}}^{\text{train}}(\theta, \lambda)$ is under the *transductive* setting, which can be easily generalized to the *inductive* setting by assuming only $\{A_{ij} \mid i, j \in V_{\text{train}}\}$ and $\{X_i \mid i \in V_{\text{train}}\}$ are used during training.

### 2.2 BACKGROUND: STANDARD DATASET CONDENSATION METHODS

Now, we review the *standard dataset condensation* (SDC) and its natural *bilevel optimization* (BL) formulation (Wang et al., 2018).

**SDC's objective.** Standard dataset condensation aims to find a synthetic graph $\mathcal{S} = (A', X', \mathbf{y}')$ of size $c \ll n$, with (weighted) adjacency matrix $A' \in \mathbb{R}_{\geq 0}^{c \times c}$, node features $X' \in \mathbb{R}^{c \times d}$, node labels $\mathbf{y}' \in [K]^c$, and (possibly) train/validation[1] splits $V'_{\text{train}} \bigcup V'_{\text{val}} = [c]$. The goal of SDC is to obtain comparable generalization performance on the real graph by training on the condensed graph, i.e., $\mathcal{L}_{\mathcal{T}}^{\text{test}}(\theta^{\mathcal{T}}, \lambda) \approx \mathcal{L}_{\mathcal{T}}^{\text{test}}(\theta^{\mathcal{S}}, \lambda)$ where $\theta^{\mathcal{S}} = arg \min_{\theta} \mathcal{L}_{\mathcal{S}}^{\text{train}}(\theta, \lambda)$ is the model parameters (of the same GNN $f_{\theta, \lambda}$) optimized on the synthetic graph. By posing $\theta^{\mathcal{S}}$ as a function of the condensed graph $\mathcal{S}$, SDC can be formulated as a *bilevel optimization* problem,

$$\mathcal{S}^* = \arg \min_{\mathcal{S}} \mathcal{L}_{\mathcal{T}}^{\text{train}}(\theta^{\mathcal{S}}(\mathcal{S}), \lambda) \quad \text{s.t.} \quad \theta^{\mathcal{S}}(\mathcal{S}) := \arg \min_{\theta} \mathcal{L}_{\mathcal{S}}^{\text{train}}(\theta, \lambda). \tag{1}$$

However, the above problem involves a nested-loop optimization and solving the inner loop for $\theta^{\mathcal{S}}(\mathcal{S})$ at each iteration requires a computationally expensive procedure: unrolling the neural network's computational graph for $\mathcal{S}$ over multiple optimization steps of $\theta$.

**SDC in a gradient matching formulation.** Zhao et al. (2020) alleviate this computational issue by introducing a *gradient matching* (GM) formulation.

*Parameter-matching formulation.* To start, we assume neural network $f_{\theta, \lambda}$ is a *locally smooth function*, and thus similar weights $\theta^{\mathcal{S}} \approx \theta^{\mathcal{T}}$ imply similar mappings. Then one can formulate the condensation objective as matching the optimized parameters (which depends on initialization $\theta_0$), i.e., finding $\mathcal{S}^* = \arg \min_{\mathcal{S}} \mathbb{E}_{\theta_0 \sim P_{\theta_0}} [D(\theta^{\mathcal{S}}(\mathcal{S}, \theta_0), \theta^{\mathcal{T}}(\theta_0))] \text{s.t.} \theta^{\mathcal{S}}(\mathcal{S}, \theta_0) := \arg \min_{\theta} \mathcal{L}_{\mathcal{S}}^{\text{train}}(\theta(\theta_0), \lambda)$, where $\theta^{\mathcal{T}}(\theta_0) := \arg \min_{\theta} \mathcal{L}_{\mathcal{T}}^{\text{train}}(\theta(\theta_0), \lambda)$ and $D(\cdot, \cdot)$ is a distance function.

*Reduction to gradient matching via approximations.* The parameter-matching problem is still a bilevel optimization but can be simplified via several approximations. *(1)* Firstly, $\theta^{\mathcal{S}}(\mathcal{S}, \theta_0)$ is approximated by the output of an incomplete gradient-descent optimization, $\theta^{\mathcal{S}}(\mathcal{S}, \theta_0) \approx \theta_{t+1}^{\mathcal{S}} \leftarrow \theta_t^{\mathcal{S}} - \eta \nabla_{\theta} \mathcal{L}_{\mathcal{S}}^{\text{train}}(\theta_t^{\mathcal{S}}, \lambda)$. However, the target parameter $\theta^{\mathcal{T}}(\theta_0)$ may be far away from $\theta_{t+1}^{\mathcal{S}}$. Zhao et al. (2020) propose to match $\theta_{t+1}^{\mathcal{S}}$ with incompletely optimized $\theta_{t+1}^{\mathcal{T}} \leftarrow \theta_t^{\mathcal{T}} - \eta \nabla_{\theta} \mathcal{L}_{\mathcal{T}}^{\text{train}}(\theta_t^{\mathcal{T}}, \lambda)$ at each iteration $t$. Consequently, the SDC's objective is now $\mathcal{S}^* = \arg \min_{\mathcal{S}} \mathbb{E}_{\theta_0 \sim P_{\theta_0}} [\sum_{t=0}^{T-1} D(\theta_t^{\mathcal{S}}, \theta_t^{\mathcal{T}})]$. *(2)* Secondly, if we assume $\theta_t^{\mathcal{S}}$ can always track $\theta_t^{\mathcal{T}}$ (i.e., $\theta_t^{\mathcal{S}} \approx \theta_t^{\mathcal{T}}$) from the initialization $\theta_0$ up to iteration $t$, then we can replace $D(\theta_{t+1}^{\mathcal{S}}, \theta_{t+1}^{\mathcal{T}})$ by $D(\nabla_{\theta} \mathcal{L}_{\mathcal{S}}^{\text{train}}(\theta_t^{\mathcal{S}}, \lambda), \nabla_{\theta} \mathcal{L}_{\mathcal{T}}^{\text{train}}(\theta_t^{\mathcal{T}}, \lambda))$ and then by this approximation $D(\nabla_{\theta} \mathcal{L}_{\mathcal{S}}^{\text{train}}(\theta_t^{\mathcal{S}}, \lambda), \nabla_{\theta} \mathcal{L}_{\mathcal{T}}^{\text{train}}(\theta_t^{\mathcal{S}}, \lambda))$. Repeating this inductive argument, the standard condensation objective is finally approximated by matching the gradients at each iteration $t$,

$$\mathcal{S}^* = \arg \min_{\mathcal{S}} \mathbb{E}_{\theta_0 \sim P_{\theta_0}} \left[ \sum_{t=0}^{T-1} D(\nabla_{\theta} \mathcal{L}_{\mathcal{S}}^{\text{train}}(\theta_t^{\mathcal{S}}, \lambda), \nabla_{\theta} \mathcal{L}_{\mathcal{T}}^{\text{train}}(\theta_t^{\mathcal{S}}, \lambda)) \right]. \tag{2}$$

With this gradient matching objective, we obtain a single deep network with parameters $\theta$ trained on the condensed graph $\mathcal{S}$. The condensed graph $\mathcal{S}$ is optimized such that the distance between the gradient vectors of $\mathcal{L}_{\mathcal{T}}^{\text{train}}$ and of $\mathcal{L}_{\mathcal{S}}^{\text{train}}$ w.r.t. the parameters $\theta$ is minimized. Cosine distance $D(\cdot, \cdot) = \cos(\cdot, \cdot)$ works well in practice (Zhao et al., 2020).

## 2.3 CHALLENGES: STANDARD DATASET CONDENSATION IS PROBLEMATIC ACROSS GNNS

For ease of theoretical discussions, in this subsection, we consider single-layer message passing GNNs. Message passing GNNs can be interpreted as iterative convolution over nodes (i.e., *message passing*) (Ding et al., 2021) where $X^{(0)} = X$, $X^{(l+1)} = \sigma(C_{\alpha^{(l)}}(A) X^{(l)} W^{(l)})$ for $l \in [L]$, and $f(A, X) = X^{(L)}$, where $C_{\alpha^{(l)}}(A)$ is the convolution matrix parametrized by $\alpha^{(l)}$, $W^{(l)}$ is the learnable linear weights, and $\sigma(\cdot)$ denotes the non-linearity. One-dimensional convolution neural networks (1D-CNNs) can be expressed by a similar formula, $f(X) = (\sum_{k=-K}^{k=K} \alpha^{(k)} P^k) XW$, parameterized by $\theta = [\alpha, W]$ where $\alpha = [\alpha^{(-K)}, \ldots, \alpha^{(K)}]$. $P$ is the cyclic permutation matrix (of a unit shift). The kernel size is $(2K + 1), K \geq 0$; see Appendix B.2 for details.

Despite the success of the gradient matching algorithm in preserving the model performance when trained on the condensed dataset (Wang et al., 2022), it naturally overfits the model $f_{\theta, \lambda}$ used during condensation and generalizes poorly to others. There is no guarantee that the condensed synthetic data $\mathcal{S}^*$ which minimizes the objective (Eq. (2)) for a specific model $f_{\theta, \lambda}$ (marked by its hyperparameter $\lambda$) can generalize well to other models $f_{\theta, \lambda'}$ where $\lambda' \neq \lambda$. We aim to demonstrate that this overfitting issue can be much more *severe on graphs* than on images, where our main theoretical results can be informally summarized as follows.

---

[1] The train/validation split of synthetic data is only required by HCDC; see Eq. (1) vs. Eq. (3).

**Proposition.** *Standard dataset condensation using gradient matching algorithm (Eq. (2)) is problematic across GNNs. The condensed graph using a single-layer message passing GNN may fail to generalize to the other GNNs with a different convolution matrix.*

We first show the successful generalization of SDC across one-dimensional *convolution neural networks* (1D-CNN). Then, we show a contrary result on GNNs: failed generalization of SDC across GNNs. These theoretical analyses demonstrate the hardness of data condensation on graphs. Our analysis is based on the achievability condition of a gradient matching objective; see Assumption 1 in Appendix A.

In Lemma 1 of Appendix C.1, under least square regression with linear GNN/CNN (see Appendix B.4 for formal definitions), if the standard dataset condensation GM objective is achievable, then the optimizer on the condensed dataset $\mathcal{S}$ is also optimal on the original dataset $\mathcal{T}$. Now, we study the generalizability of the condensed dataset across different models. We first show a successful generalization of SDC across different 1D-CNN networks; see Proposition 1 in Appendix A. As long as we use a 1D-CNN with a sufficiently large kernel size $K$ during condensation, we can generalize the condensed dataset to a wide range of models, i.e., 1D-CNNs with a kernel size $K' \leq K$.

However, we obtain a contrary result for GNNs in terms of the generalizability of condensed datasets across models. Two dominant effects, which cause the failure of the condensed graph's ability to generalize across GNNs, are discovered.

Firstly, the learned adjacency $A'$ of the synthetic graph $\mathcal{S}$ can easily *overfit* the condensation objective (see Proposition 2), and thus can fail to maintain the characteristics of the original structure and distinguish between different architectures; see Proposition 2 in Appendix A for the theoretical result and Table 1 for relevant experiments.

| Ratio ($c/n$) | $A'$ learned | $A' = I_c$ |
|---|---|---|
| 0.05% | $59.2 \pm 1.1$ | $\mathbf{61.3} \pm 0.5$ |
| 0.25% | $63.2 \pm 0.3$ | $\mathbf{64.2} \pm 0.4$ |

(a) Test accuracy of graph condensation with learned or identity adjacency.

| Condense\Test | GCN | SGC ($K=2$) | GIN |
|---|---|---|---|
| GCN | $\mathbf{60.3} \pm 0.3$ | $59.2 \pm 0.7$ | $42.2 \pm 4.3$ |
| SGC | $59.2 \pm 1.1$ | $\mathbf{60.5} \pm 0.6$ | $39.0 \pm 7.1$ |
| GIN | $47.5 \pm 3.6$ | $43.6 \pm 5.8$ | $\mathbf{59.1} \pm 1.1$ |

(b) Generalization accuracy of graphs condensed with different GNNs (row) across GNNs (column) under $c/n = 0.25\%$.

Table 1: Test accuracy of GNNs trained on condensed Ogbn-arxiv (Hu et al., 2020) graph verifying the two effects (Propositions 2 and 3) that hinders the generalization of the condensed graph across GNNs. (a) Condensed adjacency is overfitted to the SDC Objective, (b) Convolution filters and inductive bias mismatch across GNNs.

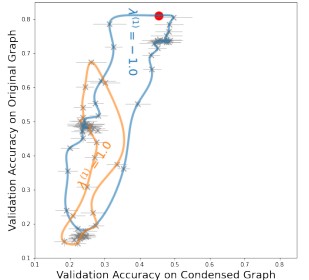

(a) Condense ratio $c/n = 0.2$

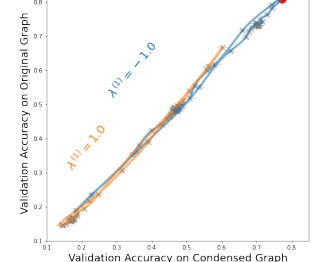

(b) Condense ratio $c/n = 0.8$

Figure 1: The manifold of GNNs with convolution filters $C_\lambda = I + \lambda^{(1)}L + \lambda^{(2)}(\frac{2}{\lambda_{\max}}L - I)$ (linear combination of first two orders of ChebNet (Defferrard et al., 2016), $\lambda$'s are hyperparameters; see Appendices B.2 and E) projected to the plane of validation accuracy on condensed (x-axis) and original (y-axis) graphs under two ratios $c/n$ on Cora (Yang et al., 2016). The GNN with $C = (\frac{2}{\lambda_{\max}} - 1)L \propto L$ (red dot) is a biased point in this model space.

Secondly, GNNs differ from each other mostly on the design of convolution filter $C(A)$, i.e., how the convolution weights $C$ depend on the adjacency information $A$. The convolution filter $C(A)$ used during condensation is a single biased point in "the space of convolutions"; see Fig. 1 for a visualization, thus there is a *mismatch* of inductive bias when transferring to a different GNN. These two effects lead to the obstacle when transferring the condensed graph across GNNs, which is formally characterized by Proposition 3 in Appendix A.

Proposition 3 provides an effective lower-bound on the relative estimation error of optimal model parameters when a different convolution filter $C'(\cdot) \neq C(\cdot)$ is used.[2] According to the spectral characterization of convolution filters of GNNs (Table 1 of (Balcilar et al., 2021)), we can approximately compute the maximum eigenvalue of $Q$ for some GNNs. For example, if we condense with $f^C$ graph

---

[2]If $C'(\cdot) = C(\cdot)$ Lemma 1 guarantees $W_{C'}^{\mathcal{S}} = W_{C'}^{\mathcal{T}}$ and the lower bound in Proposition 3 is 0.

isomorphism network (GIN-0) (Xu et al., 2018) but train $f^{C'}$ GCN on the condensed graph, we have $\|W_{C'}^{\mathcal{S}} - W_{C'}^{\mathcal{T}}\|/\|W_{C'}^{\mathcal{T}}\| \gtrsim \overline{\deg} + 1$ where $\overline{\deg}$ is the average node degree of the original graph. This large lower bound hints the catastrophic failure when transferring across GIN and GCN; see Table 1.

## 3 HYPERPARAMETER CALIBRATED DATASET CONDENSATION

Our goal is to develop an optimal and reliable condensation method for architecture/hyperparameter search. Standard dataset condensation objective (Eq. (1)/Eq. (2)) does not accomplish this goal since it does not generalize across GNNs, as proven in Section 2.3. In this section, we propose a new condensation objective specifically for *preserving the outcome of hyperparameter optimization* (HPO) on the condensed dataset.

**HPO's objective.** HPO finds the optimal hyperparameter $\lambda^{\mathcal{T}}$ such that the corresponding model $f_{\theta,\lambda^{\mathcal{T}}}$ minimizes the validation loss after training, i.e.,

$$\lambda^{\mathcal{T}} = \arg\min_{\lambda \in \Lambda} \mathcal{L}_{\mathcal{T}}^*(\lambda) \text{ where } \mathcal{L}_{\mathcal{T}}^*(\lambda) := \mathcal{L}_{\mathcal{T}}^{\text{val}}(\theta^{\mathcal{T}}(\lambda), \lambda) \text{ and } \theta^{\mathcal{T}}(\lambda) := \arg\min_{\theta} \mathcal{L}_{\mathcal{T}}^{\text{train}}(\theta, \lambda). \quad \text{(HPO)}$$

We see HPO itself is a bilevel optimization, where the optimal parameter $\theta^{\mathcal{T}}(\lambda)$ is posed as a function of the hyperparameter $\lambda$, and so is the optimized validation loss $\mathcal{L}_{\mathcal{T}}^*(\lambda)$; see Fig. 3 for illustration.

**Dataset condensation for HPO.** If both the train and validation sets are defined on the condensed dataset $\mathcal{S}$, the optimal hyperparameter $\lambda^{\mathcal{S}}$ is well-defined. Our goal is to find the synthetic dataset $\mathcal{S}$ such that we can obtain *comparable validation performance* if the hyperparameters are optimized on the condensed dataset, i.e., $\mathcal{L}_{\mathcal{T}}^*(\lambda^{\mathcal{T}}) \approx \mathcal{L}_{\mathcal{T}}^*(\lambda^{\mathcal{S}})$. Clearly, this goal looks very similar to the goal of standard dataset condensation, preserving generalization performance $\mathcal{L}_{\mathcal{T}}^{test}(\theta^{\mathcal{T}}) \approx \mathcal{L}_{\mathcal{T}}^{test}(\theta^{\mathcal{S}})$, which inspires us to formulate the new objective as a bilevel optimization problem too,

$$\mathcal{S}^* = \arg\min_{\mathcal{S}} \mathcal{L}_{\mathcal{T}}^*\big(\lambda^{\mathcal{S}}(\mathcal{S})\big) \quad \text{s.t.} \quad \lambda^{\mathcal{S}}(\mathcal{S}) = \arg\min_{\lambda \in \Lambda} \mathcal{L}_{\mathcal{S}}^*(\lambda), \tag{3}$$

where optimized validation losses $\mathcal{L}_{\mathcal{T}}^*(\cdot)$ and $\mathcal{L}_{\mathcal{S}}^*(\cdot)$ are defined following Eq. (HPO).

However, two challenges exist: (1) This formulation (Eq. (3)) is a nested optimization (for dataset condensation) over another nested optimization (for HPO) which is challenging to solve as high order gradients are required. (2) In addition, another challenge lies in the search space/feasible set of the hyperparameters $\Lambda$. In contrast to parameter optimization, where the search space is usually assumed to be the continuous and unbounded Euclidean space, the search space of hyperparameters $\Lambda$ can be either a discrete set or a continuous one. Examples of discrete hyperparameters include neural network type, width, depth, batch size, etc. Often we face compositions of these discrete- and continuous-natured hyperparameters, and we can either model them all as discrete ones and search by grid search, Bayesian optimization, and reinforcement learning; or relax the discrete search space to a continuous one.

**Hyperparameter calibration: a sufficient alternative to HPO's objective.** To solve the aforementioned two challenges, we propose an sufficient alternative to Eq. (HPO). Specifically, we propose to identify the condensed dataset that preserves the outcome of HPO on $\Lambda$ without solving the HPO objective. We call this hyperparameter calibration, which is formally defined in Definition 1.

**Definition 1** (Hyperparameter Calibration). *Given original dataset $\mathcal{T}$, generic model $f_{\theta,\lambda}$, and hyperparameter search space $\Lambda$, we say a condensed dataset $\mathcal{S}$ is hyperparameter calibrated, if for any $\lambda_1 \neq \lambda_2 \in \Lambda$, it holds that,*

$$\big(\mathcal{L}_{\mathcal{T}}^*(\lambda_1) - \mathcal{L}_{\mathcal{T}}^*(\lambda_2)\big)\big(\mathcal{L}_{\mathcal{S}}^*(\lambda_1) - \mathcal{L}_{\mathcal{S}}^*(\lambda_2)\big) > 0, \tag{HC}$$

*In other words, changes of the optimized validation loss on $\mathcal{T}$ and $\mathcal{S}$ always have the same sign between hyperparameters $\lambda_1 \neq \lambda_2$.*

It is clear that if hyperparameter calibration (HC) is satisfied, HPO on the original and condensed datasets yields the same result. Therefore, our mission changes to *ensuring hyperparameter calibration for a single pair of hyperparameters* $(\lambda_1, \lambda_2)$.

**HCDC: hypergradient alignment objective for dataset condensation.** To proceed, we assumes the existence of *a continuous extension of the search space*: the (possibly discrete) search space $\Lambda$ can be extended to a compact and connected set $\tilde{\Lambda} \supset \Lambda$, where we can define continuation of the

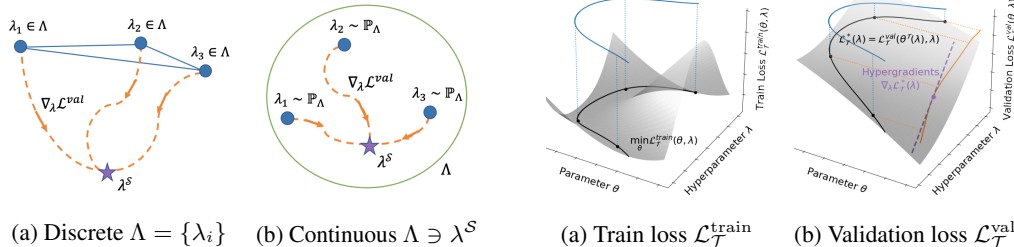

(a) Discrete $\Lambda = \{\lambda_i\}$    (b) Continuous $\Lambda \ni \lambda^{\mathcal{S}}$     (a) Train loss $\mathcal{L}_{\mathcal{T}}^{\mathrm{train}}$    (b) Validation loss $\mathcal{L}_{\mathcal{T}}^{\mathrm{val}}$

Figure 2: Illustration of the constructed extended search space $\tilde{\Lambda}$ illustrated as the orange trajectory for both **a)** discrete $\Lambda$ and **(b)** continuous $\Lambda$. The trajectory starts from $\lambda_{i,0}^{\mathcal{S}} = \lambda_i \in \Lambda$ for discrete $\Lambda$ (or random points for continuous $\Lambda$), and updates through $\lambda_{i,t+1}^{\mathcal{S}} \leftarrow \lambda_{i,t}^{\mathcal{S}} - \eta \nabla_\lambda \mathcal{L}_{\mathcal{S}}^*(\lambda_{i,t}^{\mathcal{S}})$.

Figure 3: Loss landscape w.r.t. $\theta$ and $\lambda$. A hyperparameter $\lambda$ has an optimal parameter $\theta^{\mathcal{T}}(\lambda)$ (blue curve in $(\theta, \lambda)$-plane in **(a)**) that minimizes the train loss. In **(b)**, injecting optimal parameters $\theta^{\mathcal{T}}(\lambda)$ into the validation loss, we obtain a function of validation loss w.r.t. $\lambda$ (denoted as $\mathcal{L}_{\mathcal{T}}^*(\lambda)$) in $(\mathcal{L}, \lambda)$-plane, shown as the orange curve. The purple dash line illustrates the hypergradients, i.e., gradient of $\mathcal{L}_{\mathcal{T}}^*(\lambda)$ w.r.t. $\lambda$.

generic model $f_{\theta,\lambda}$ on $\tilde{\Lambda}$ so that $f_{\theta,\lambda}$ is differentiable anywhere in $\tilde{\Lambda}$. In Section 4, we will elaborate on how to construct such an extended search space $\tilde{\Lambda}$.

Now, with the existence of such a continual extension of the search space, if we limit our step size to be small, we only need to ensure hyperparameter calibration in Eq. (HC) under the special case that $\lambda_1$ is within the neighborhood of $\lambda_2$, i.e., $\lambda_1 \in B_r(\lambda_2)$ for some $r > 0$. The change in validation loss is approximated *up to first-order* by the hypergradients, i.e., $\mathcal{L}_{\mathcal{T}}^*(\lambda_1) - \mathcal{L}_{\mathcal{T}}^*(\lambda_2) \approx \langle \nabla_\lambda \mathcal{L}_{\mathcal{T}}^*(\lambda), \Delta\lambda \rangle$, where $\lambda_1 = \lambda + \Delta\lambda$, $\lambda_2 = \lambda$ with $r \geq \|\Delta\lambda\|_2 \to 0^+$. The hyperparameter calibration condition within this tiny neighborhood $B_r(\lambda)$ is then simplified to $\nabla_\lambda \mathcal{L}_{\mathcal{T}}^*(\lambda) \parallel \nabla_\lambda \mathcal{L}_{\mathcal{S}}^*(\lambda)$, i.e., the two hypergradient vectors are aligned and pointing to the same direction.

Assuming the extended search space $\tilde{\Lambda}$ can be covered by the union of many small neighborhoods, we derive the following notion and equivalence relation of hypergradient alignment.

**Definition 2** (Hypergradient Alignment). *We say hypergradients are aligned in an extended search space $\tilde{\Lambda}$, if for any $\lambda \in \tilde{\Lambda}$, it holds that $\nabla_\lambda \mathcal{L}_{\mathcal{T}}^*(\lambda) \parallel \nabla_\lambda \mathcal{L}_{\mathcal{S}}^*(\lambda)$, i.e., $\cos(\nabla_\lambda \mathcal{L}_{\mathcal{T}}^*(\lambda), \nabla_\lambda \mathcal{L}_{\mathcal{S}}^*(\lambda)) = 0$.*

**Theorem 1** (Equivalence between Hypergradient Alignment and Hyperparameter Calibration). *Hypergradient alignment (Definition 2) is equivalent to hyperparameter calibration (Definition 1) on the connected and compact set, e.g., the extended search space $\tilde{\Lambda}$.*

We summarize the relations between the two notions (Definitions 1 and 2) as follows,

Hypergrad. Alignment in $\tilde{\Lambda}$ $\Longleftrightarrow$ Hyperpara. Calibration in $\tilde{\Lambda}$ $\Longrightarrow$ Hyperpara. Calibration in $\Lambda$.

Therefore, hypergradient alignment on $\tilde{\Lambda}$ is sufficient to ensure hyperparameter calibration on $\Lambda$, and hence the outcome of HPO over $\Lambda$ is preserved. Consequently, as the core of our hyperparameter calibrated dataset condensation (HCDC), we propose the hypergradient alignment objective below

$$\mathcal{S}^* = \arg\min_{\mathcal{S}} \sum_{\lambda \in \tilde{\Lambda}} D\big(\nabla_\lambda \mathcal{L}_{\mathcal{T}}^{\mathrm{val}}(\theta^{\mathcal{T}}(\lambda), \lambda), \nabla_\lambda \mathcal{L}_{\mathcal{S}}^{\mathrm{val}}(\theta^{\mathcal{S}}(\lambda), \lambda)\big), \tag{HCDC}$$

where cosine distance $D(\cdot, \cdot) = \cos(\cdot, \cdot)$ is used.

## 4 IMPLEMENTATIONS OF HCDC AND APPLICATIONS TO GNNS

Finally, we work on implementing and simplifying the hyperparameter calibrated dataset condensation (HCDC) objective and apply it to the graph architecture/hyperparameter search problem.

**Constructing the extended search space $\tilde{\Lambda}$.** The HCDC objective requires hypergradient alignment over all $\lambda$'s in an extended space $\tilde{\Lambda}$ that is a compact and connected superset of $\Lambda$. Under the discrete search space $\Lambda$, which consists of $p$ candidate hyperparameters, one can naively construct $\tilde{\Lambda}$ as $O(p^2)$ continuous paths connecting pairs of candidate hyperparameters (shown as blue lines in Fig. 2a). This is apparently undesirable due to its quadratic complexity in $p$. We propose a construction of $\tilde{\Lambda}$ with a linear complexity in $p$, which works as follows. For any $i \in [p]$, we construct a "representative" path, named $i$-th HPO trajectory, which starts from $\lambda_{i,0}^{\mathcal{S}} = \lambda_i \in \Lambda$ and

updates through $\lambda_{i,t+1}^{\mathcal{S}} \leftarrow \lambda_{i,t}^{\mathcal{S}} - \eta \nabla_\lambda \mathcal{L}_{\mathcal{S}}^*(\lambda_{i,t}^{\mathcal{S}})$, shown as the orange dashed lines in Fig. 2a. All of the $p$ trajectories will approach the optima $\lambda^{\mathcal{S}}$, forming "connected" paths between any pair of hyperparameters $\lambda_i \neq \lambda_j \in \Lambda$. This construction is also used in a continuous search space to save computation (except that we randomly select the starting points $\lambda_i \sim \mathbb{P}_\Lambda$). To extend general discrete neural architecture space $\Lambda$ into a continuously differentiable $\tilde{\Lambda}$, differentiable NAS approaches surveyed in Appendix D.3 can be used, and we leave exploration in this direction to future work.

**Computing hypergradients and optimizing hypergradient alignment loss in Eq. (HCDC).** The hypergradients are the gradients of the optimized validation loss $\mathcal{L}_{\mathcal{T}}^*(\lambda) = \mathcal{L}_{\mathcal{T}}^{\text{val}}(\theta^{\mathcal{T}}(\lambda), \lambda)$ w.r.t the hyperparameters $\lambda$; see Fig. 3 for the illustration. The efficient computation of hypergradients $\nabla_\lambda \mathcal{L}_{\mathcal{T}}^*(\lambda)$ and $\nabla_\lambda \mathcal{L}_{\mathcal{S}}^*(\lambda)$ uses the implicit function theorem (IFT),

$$\nabla_\lambda \mathcal{L}_{\mathcal{T}}^*(\lambda) = -\left[\frac{\partial^2 \mathcal{L}_{\mathcal{T}}^{\text{train}}(\theta,\lambda)}{\partial \lambda \partial \theta^T}\right]\left[\frac{\partial^2 \mathcal{L}_{\mathcal{T}}^{\text{train}}(\theta,\lambda)}{\partial \theta \partial \theta^T}\right]^{-1} \nabla_\theta \mathcal{L}_{\mathcal{T}}^{\text{val}}(\theta,\lambda) + \nabla_\lambda \mathcal{L}_{\mathcal{T}}^{\text{val}}(\theta,\lambda), \qquad \text{(IFT)}$$

where $\nabla_\lambda \mathcal{L}_{\mathcal{T}}^{\text{val}}(\theta,\lambda)$ is the direct gradient, which is 0 when $\lambda$ only affects the loss through the model $f_{\theta,\lambda}$. The first term is the product of the training mixed partials $\left[\frac{\partial^2 \mathcal{L}_{\mathcal{T}}^{\text{train}}(\theta,\lambda)}{\partial \lambda \partial \theta^T}\right]$, inverse training Hessian $\left[\frac{\partial^2 \mathcal{L}_{\mathcal{T}}^{\text{train}}(\theta,\lambda)}{\partial \theta \partial \theta^T}\right]^{-1}$, and the validation gradients $\nabla_\theta \mathcal{L}_{\mathcal{T}}^{\text{val}}(\theta,\lambda)$. While the other parts can be computed by back-propagation, the inverse Hessian needs to be approximated. Instead of using the conjugate gradient method, Lorraine et al. (2020) propose a stable, tractable and efficient Neumann series approximation, $\left[\frac{\partial^2 \mathcal{L}_{\mathcal{T}}^{\text{train}}(\theta,\lambda)}{\partial \theta \partial \theta^T}\right]^{-1} = \lim_{i \to \infty} \sum_{j=0}^{i} \left[I - \frac{\partial^2 \mathcal{L}_{\mathcal{T}}^{\text{train}}(\theta,\lambda)}{\partial \theta \partial \theta^T}\right]^j$ with constant memory constraint. To optimize the synthetic validation set $\mathcal{S}^{\text{val}}$ w.r.t. the cosine hypergradient matching loss in Eq. (HCDC), we only need to take gradients of $\nabla_\theta \mathcal{L}_{\mathcal{S}}^{\text{val}}(\theta,\lambda)$ and $\nabla_\lambda \mathcal{L}_{\mathcal{S}}^{\text{val}}(\theta,\lambda)$ w.r.t. $\mathcal{S}^{\text{val}}$, which can be handled by the same back-propagation technique in SDC, where we take gradients of $\nabla_\theta \mathcal{L}_{\mathcal{S}}^{\text{train}}(\theta,\lambda)$ w.r.t $\mathcal{S}^{\text{train}}$.

**Connecting HCDC to SDC (Eq. (2)).** Theoretically speaking, the objective of HCDC, preserving the outcome of hyperparameter optimization (HPO), is orthogonal to the objective of SDC, which preserves the generalization performance. Therefore, we use SDC to learn the synthetic training dataset $\mathcal{S}^{\text{train}}$ in Eq. (2) and HCDC to learn the synthetic validation dataset $\mathcal{S}^{\text{val}}$ in Eq. (HCDC). Learning the synthetic training and validation dataset may result in disconnected training and validation set, which is allowed in graph learning.

**Which graph architecture/hyperparameter search problems can HCDC solve?** We illustrate how to tackle the two types of search spaces: (1) discrete and finite $\Lambda$ and (2) continuous and bounded $\Lambda$ with two typical examples originated from the problem of searching for the best convolution matrix $C(A)$ on a large graph $\mathcal{T} = (A, X, \mathbf{y})$. **(1)** *Discrete and finite search space $\Lambda$*: often the most important question of architecture search on large graphs is *what design of convolution filter performs best on the given graph*? One may simply train the set of $p$ prior-defined GNNs $\{f_{[\alpha^{(i)}, W]}^{C^{(i)}} \mid i = 1, \ldots, p\}$ whose convolution matrices are $\mathcal{C} = \{C_{\alpha^{(1)}}^{(1)}(A), \ldots, C_{\alpha^{(p)}}^{(p)}(A)\}$ and compare their validation performance. We can formulate this problem as HPO, by defining an "interpolated" model $f_{[\alpha, W], \lambda}^{\mathcal{C}}$ whose convolution matrix is $C_{\alpha, \lambda}(A) = \lambda^{(1)} C_{\alpha^{(1)}}^{(1)}(A) + \cdots + \lambda^{(p)} C_{\alpha^{(p)}}^{(p)}(A)$, where hyperparameters $\lambda = [\lambda^{(1)}, \ldots, \lambda^{(p)}] \in \Lambda$ and parameters $\alpha = [\alpha^{(1)}, \ldots, \alpha^{(p)}]$. The search space $\Lambda = \{\lambda_1 = \mathbf{e}_1^p, \ldots, \lambda_p = \mathbf{e}_p^p\}$ is the set of unit vectors in $\mathbb{R}^p$. **(2)** *Continuous and bounded search space $\Lambda$*: one may also use a continuous generic formula, e.g., truncated series, to model a wide range of convolution filters, i.e., $C_\lambda(A) = \sum_{i=1}^{p} \lambda^{(i)} C^{(i)}(A)$, for example in ChebNet (Defferrard et al., 2016) or SIGN (Frasca et al., 2020) (see Appendix B.2). The formula of $C_\lambda(A)$ is a special case of the $C_{\alpha, \lambda}(A)$ in (1), despite the search space $\Lambda$ is now continuous.

**The complete pseudo-code of HCDC.** We conclude this section by summarizing the implementation of HCDC in Algorithm 1. We assume a discrete and finite search space $\Lambda$.

In Line 8, to compute $\nabla_{\mathcal{S}_{\text{val}}} D\big(\nabla_\lambda \mathcal{L}_{\mathcal{T}}^*(\lambda), \nabla_\lambda \mathcal{L}_{\mathcal{S}}^*(\lambda)\big)$, we note that only $\nabla_\lambda \mathcal{L}_{\mathcal{S}}^*(\lambda)$ depends on $\mathcal{S}_{\text{val}}$. By Eq. (IFT), $\nabla_\lambda \mathcal{L}_{\mathcal{S}}^*(\lambda) = -\left[\frac{\partial^2 \mathcal{L}_{\mathcal{S}}^{\text{train}}(\theta,\lambda)}{\partial \lambda \partial \theta^S}\right]\left[\frac{\partial^2 \mathcal{L}_{\mathcal{S}}^{\text{train}}(\theta,\lambda)}{\partial \theta \partial \theta^S}\right]^{-1} \nabla_\theta \mathcal{L}_{\mathcal{S}}^{\text{val}}(\theta,\lambda)$ (there is no direct gradients since $\lambda$ only affects the loss through the model $f_{\theta,\lambda}$). Since only the validation loss term $\nabla_\theta \mathcal{L}_{\mathcal{S}}^{\text{val}}(\theta,\lambda)$ depends on $\mathcal{S}_{\text{val}}$, we only need to compute $\nabla_{\mathcal{S}_{\text{val}}} \nabla_\theta \mathcal{L}_{\mathcal{S}}^{\text{val}}(\theta,\lambda)$ by back-propagation.

---

**Algorithm 1** HCDC: hyperparameter calibrated dataset condensation, which aims to preserve the validation performance ranking of architectures/hyperparameters.

---

**Require:** Original dataset $\mathcal{T}$. A set of NN architectures $f_{\theta,\lambda}$ where $\lambda \in \Lambda = \{\lambda_1, \ldots, \lambda_p\}$.
**Require:** Condensed training data $\mathcal{S}_{\text{train}}$ learned by standard gradient-matching algorithm (Eq. (2)). Randomly initialized synthetic graph $\mathcal{S}_{\text{val}}$ for $C$ classes.
1  **for** repeat $k = 0, \ldots, K - 1$ **do**
2     **for** $\lambda = \lambda_1, \ldots, \lambda_p$ **do**
3        Initialize $\theta \leftarrow \theta_0 \sim P_{\theta_0}$
4        **for** epoch $t = 0, \ldots, T_\theta - 1$ **do**
5           Update $\theta \leftarrow \theta - \eta_\theta \nabla_\theta \mathcal{L}_{\mathcal{S}}^{\text{train}}(\theta, \lambda)$.
6           **if** $t \mod T_\lambda = 0$ **then**
7              Update $\lambda \leftarrow \lambda - \eta_\lambda \nabla_\lambda \mathcal{L}_{\mathcal{S}}^*(\lambda)$.       ▷ *Hypergradients calculated using Eq.* (IFT).
8           Update $\mathcal{S}_{\text{val}} \leftarrow \mathcal{S}_{\text{val}} - \eta_{\mathcal{S}} \nabla_{\mathcal{S}_{\text{val}}} D\big(\nabla_\lambda \mathcal{L}_{\mathcal{T}}^*(\lambda), \nabla_\lambda \mathcal{L}_{\mathcal{S}}^*(\lambda)\big)$
9  **return** Condensed validation data $\mathcal{S}_{\text{val}}$.

---

## 5   Related Work

**Graph condensation** (Jin et al., 2021) achieved the state-of-the-art on preserving GNNs' performance on the simplified graph. Jin et al. (2021) adapted the gradient matching algorithm (Zhao et al., 2020) (Eq. (2)) to graph data, together with a MLP-based graph generative model (Anand & Huang, 2018), leaving out several major issues on its efficiency, performance, and generalizability. While the efficiency was improved by reducing the number of gradient matching steps (Jin et al., 2022), the performance degradation on medium- and large-sized graphs still renders graph condensation impractical. Our HCDC is designed for hyperparameter/architecture search, where we train multiple models on the same dataset and the efficiency gain is much more significant.

**Implicit differentiation** methods apply the implicit function theorem (IFT) (Eq. (IFT)) to nested-optimization problems (Wang et al., 2019). Lorraine et al. (2020) approximated the inverse Hessian by Neumann series, which is a stable alternative to conjugate gradients (Shaban et al., 2019) and scales IFT to large networks with constant memory. Lorraine et al. (2020) also showed that unrolling differentiation around locally optimal parameters for $i$ steps is equivalent to approximating the inverse Hessian by Neumann series up to the first $i$ terms.

In addition, we summarize graph reduction methods (including graph coreset selection, graph sampling, graph sparsification, and graph coarsening), as well as more dataset condensation and coreset selection methods beyond graphs and differentiable NAS methods in Appendix D.

## 6   Experiments

In this section we validate the effectiveness of hyperparameter calibrated dataset condensation (HCDC) when applied to speed up graph architecture/hyperparameter search. Spearman's rank correlation coefficient $r_s$ between two rankings of the ordered list of hyperparameters on the original and condensed datasets, which is concisely referred to as correlation, is used as an important evaluation metric, in addition to the percentage accuracy metric (referred to as performance).

**Synthetic experiments on CIFAR-10.** We first consider a synthetically created set of hyperparameters on an image dataset, CIFAR-10. Consider the $M$-fold cross validation, where a fraction of $1/M$ samples are used as the validation dataset each time. The $M$-fold cross-validation process can be modeled by a set of $M$ hyperparameters $\{\varphi_i \in \{0, 1\} \mid i = 1, \ldots, M\}$, where $\varphi_i = 1$ if and only if the $i$-th fold is used for validation. The problem of finding the best validation performance among the $M$ results can be modeled as a hyperparameter optimization problem with a discrete search space $|\Lambda| = M$. We compare HCDC with the gradient matching (Zhao et al., 2020) and distribution matching (Zhao & Bilen, 2021b) baselines. We also consider a uniform random sampling baseline and an early-stopping baseline where we train the same number of *iterations* (with the same batchsize) as the other methods but on the original dataset. The results of $M = 20$ and $c/n = 2\%$ and $4\%$ are reported in Table 2, where we see HCDC achieves the highest rank correlation. This experiment shows that HCDC can be applied to general types of data and tasks as long as the extended search space can be effectively and efficiently constructed.

| | Ratio ($c_{train}/n$) | |
|---|---|---|
| Method | 2% | 4% |
| Random | −0.03 | 0.07 |
| SDC-GM | 0.64 | 0.78 |
| SDC-DM | 0.77 | 0.86 |
| Early-Stopping | 0.11 | 0.24 |
| HCDC | 0.91 | 0.94 |

Table 2: The rank correlation and validation performance on the original dataset of the $M$-fold cross validation ranked/selected on the condensed dataset on CIFAR-10.

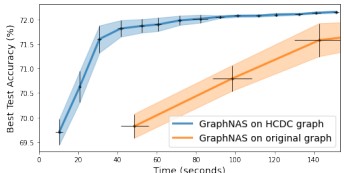

Figure 4: Speed-up of the search process of graph NAS when combined with HCDC on Ogbn-arxiv, best test performance so far vs. time spent.

**Finding the best convolution filter on (large) graphs.** One application of HCDC we analyzed in Section 4 is to speed up the selection of the best-suited convolution filter design on large graphs. Following the method discussed in Section 4, we test HCDC against (1) Random: the random uniform sampling of nodes and find their induced subgraph, (2) GCond-X: graph condensation (Jin et al., 2021) but fix the synthetic adjacency to identity, (3) GCond: graph condensation algorithm in (Jin et al., 2021), and (4) Whole Graph: when the model selection is performed on the original dataset. We use random uniform sampling to find the training synthetic subgraph before we apply HCDC. For the other coreset/condensation methods, which do not define the validation split, we randomly split the train and validation nodes according to the original split ratio. We report not only the Spearman's rank correlation, but also the test performance (on the original dataset) of the model selected by the condensed dataset. In Table 3, we see HCDC consistently outperforms the other approaches, and the test performance of selected architecture is close to the ground-truth best performance.

| Method | Ratio | Random | | GCond-X | | GCond | | HCDC | | Whole Graph |
|---|---|---|---|---|---|---|---|---|---|---|
| Dataset | ($c_{train}/n$) | Corr. | Perf. (%) | Corr. | Perf. (%) | Corr. | Perf. (%) | Corr. | Perf. (%) | Perf. (%) |
| Cora | 0.9% | $0.29 \pm .08$ | $81.2 \pm 1.1$ | $0.16 \pm .07$ | $79.5 \pm 0.7$ | $0.61 \pm .03$ | $81.9 \pm 1.6$ | $\mathbf{0.80 \pm .03}$ | $\mathbf{83.0 \pm 0.2}$ | |
| | 1.8% | $0.40 \pm .04$ | $81.9 \pm 0.5$ | $0.21 \pm .07$ | $80.3 \pm 0.4$ | $0.76 \pm .06$ | $83.2 \pm 0.9$ | $\mathbf{0.85 \pm .03}$ | $\mathbf{83.4 \pm 0.2}$ | $83.8 \pm 0.4$ |
| | 3.6% | $0.51 \pm .04$ | $82.2 \pm 0.6$ | $0.23 \pm .04$ | $80.9 \pm 0.6$ | $0.81 \pm .04$ | $83.2 \pm 1.1$ | $\mathbf{0.90 \pm .01}$ | $\mathbf{83.4 \pm 0.3}$ | |
| Citeseer | 1.3% | $0.38 \pm .11$ | $71.9 \pm 0.8$ | $0.15 \pm .07$ | $70.7 \pm 0.9$ | $0.68 \pm .03$ | $71.3 \pm 1.2$ | $\mathbf{0.79 \pm .01}$ | $\mathbf{73.1 \pm 0.2}$ | |
| | 2.6% | $0.56 \pm .06$ | $72.2 \pm 0.4$ | $0.29 \pm .05$ | $70.8 \pm 0.5$ | $0.79 \pm .05$ | $71.5 \pm 0.7$ | $\mathbf{0.83 \pm .02}$ | $\mathbf{73.3 \pm 0.5}$ | $73.7 \pm 0.6$ |
| | 5.2% | $0.71 \pm .05$ | $73.0 \pm 0.3$ | $0.35 \pm .08$ | $70.2 \pm 0.4$ | $0.83 \pm .03$ | $71.1 \pm 0.8$ | $\mathbf{0.89 \pm .02}$ | $\mathbf{73.4 \pm 0.4}$ | |
| Ogbn-arxiv | 0.1% | $0.59 \pm .08$ | $70.1 \pm 1.7$ | $0.39 \pm .06$ | $69.8 \pm 1.4$ | $0.59 \pm .07$ | $70.3 \pm 1.4$ | $\mathbf{0.77 \pm .04}$ | $\mathbf{71.9 \pm 0.8}$ | |
| | 0.25% | $0.63 \pm .05$ | $70.3 \pm 1.3$ | $0.44 \pm .03$ | $70.1 \pm 0.7$ | $0.64 \pm .05$ | $70.5 \pm 1.0$ | $\mathbf{0.83 \pm .03}$ | $\mathbf{72.4 \pm 1.0}$ | $73.2 \pm 0.8$ |
| | 0.5% | $0.68 \pm .07$ | $70.9 \pm 1.0$ | $0.47 \pm .05$ | $70.0 \pm 0.7$ | $0.67 \pm .05$ | $71.1 \pm 0.6$ | $\mathbf{0.88 \pm .03}$ | $\mathbf{72.6 \pm 0.6}$ | |
| Reddit | 0.1% | $0.42 \pm .09$ | $92.1 \pm 1.6$ | $0.39 \pm .04$ | $90.9 \pm 0.8$ | $0.53 \pm .06$ | $90.9 \pm 1.7$ | $\mathbf{0.79 \pm .03}$ | $\mathbf{92.1 \pm 0.9}$ | |
| | 0.25% | $0.50 \pm .06$ | $92.7 \pm 1.3$ | $0.41 \pm .05$ | $90.9 \pm 0.5$ | $0.61 \pm .04$ | $91.2 \pm 1.2$ | $\mathbf{0.83 \pm .01}$ | $\mathbf{92.9 \pm 0.7}$ | $94.1 \pm 0.7$ |
| | 0.5% | $0.58 \pm .06$ | $92.8 \pm 0.7$ | $0.42 \pm .03$ | $91.5 \pm 0.6$ | $0.66 \pm .02$ | $92.1 \pm 0.9$ | $\mathbf{0.87 \pm .01}$ | $\mathbf{93.1 \pm 0.5}$ | |

Table 3: Spearman's rank correlation and test performance of convolution filter selected on the condensed graph.

**Speeding up off-the-shelf graph architecture search algorithms.** Finally, we test HCDC on how much speed-up it can provide to the off-the-shelf graph architecture search methods. We use graph NAS (Gao et al., 2019) on Ogbn-arxiv with a condensation ratio of $c/n = 0.5\%$. The search space of architectures is the same as the set used in Table 3 with a focus on graphs with different convolution filters. We plot the best test performance of searched architecture (so far) versus the time spent during searching (in seconds) in Fig. 4. We see HCDC, as a dataset condensation approach, can further speed up the search process of graph NAS and is orthogonal to the efficient search algorithms like Bayesian optimization or reinforcement learning used by NAS methods.

## 7 CONCLUSION

This paper considers a novel objective for dataset condensation: preserving the outcome of hyperparameter search/optimization. We propose the hyperparameter calibration formulation for this goal, which is then realized by aligning the hyperparameter gradients. We demonstrate both theoretically and experimentally that HCDC can effectively preserve the validation performance rankings of GNNs and accelerate the hyperparameter/architecture search on graphs. However, the overall performance of HCDC can be affected by (1) how the supernet generalize to unseen architectures; (2) where we align hypergradients in the search space; (3) how we learn the synthetic training set; (4) how we parameterize the synthetic graph/dataset; and leave the heuristic exploration of all possible techniques for these design choices for future work. Beyond graph datasets, HCDC has the potential to be integrated with differentiable neural architecture search (NAS) methods (Liu et al., 2018; Wang et al., 2020) to address general neural architecture space. We hope our work opens up a promising new avenue for speeding up hyperparameter/architecture search by compressing the underlying dataset.

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

# Supplementary Material

## A    STANDARD DATASET CONDENSATION IS PROBLEMATIC ACROSS GNNS

In this section, we complete the theoretical details behind Section 2.3, which shows standard dataset condensation is problematic across GNNs.

**Assumption 1** (Achievability of a gradient matching Objective). *A gradient matching objective is defined to be achievable if there exists a non-degenerate trajectory* $(\theta_t^{\mathcal{S}})_{t=0}^{T-1}$ *(i.e., a trajectory that spans the entire parameter space* $\Theta$, *i.e.,* $\mathrm{span}(\theta_0^{\mathcal{S}}, \ldots, \theta_{T-1}^{\mathcal{S}}) \supseteq \Theta$*), such that the gradient matching loss (the objective of Eq. (2) without expectation) on this trajectory is* $0$.

**Proposition 1** (Successful Generalization of SDC across 1D-CNNs). *Consider least-squares regression with one-dimensional linear convolution* $f^{2K+1}(X)_\theta = (\sum_{k=-K}^{k=K} \alpha^{(k)} P^k) X W$ *parameterized by* $\theta = [\alpha, W]$ *where* $\alpha = [\alpha^{(-K)}, \ldots, \alpha^{(K)}]$. *P is the cyclic permutation matrix (of a unit shift). The kernel size is* $(2K+1), K \geq 0$. *If the gradient matching objective of* $f^{2K+1}$ *is achievable, then the condensed dataset* $\mathcal{S}^*$ *achieves the gradient matching objective on any trajectory* $\{\theta_t^{\prime \mathcal{S}}\}_{t=0}^{T-1}$ *for any linear convolution* $f_{\theta'}^{2K'+1}$ *with kernel size* $(2K'+1), K \geq K' \geq 0$.

The intuition behind Proposition 1 is that the 1D-CNN of kernel size $(2K+1)$ is a "supernet" of the 1D-CNN of kernel size $(2K'+1)$ if $K' \leq K$, and the condensed dataset via a bigger model can generalize well to smaller ones. This result suggests us to *use a sufficiently large model during condensation*, to enable the generalization of the condensed dataset to a wider range of models.

**Proposition 2** (Condensed Adjacency Overfits SDC Objective). *Consider least-squares regression with a linear GNN,* $f(A, X) = C(A)XW$ *parameterized by* $W$ *and* $C(A)$ *which depends on graph adjacency* $A$. *For any (full-ranked) synthetic node features* $X' \in \mathbb{R}^{c \times d}$, *there exists a synthetic adjacency matrix* $A' \in \mathbb{R}_{\geq 0}^{c \times c}$ *such that the gradient matching objective is achievable.*

**Proposition 3** (Failed Generalization of SDC across GNNs). *Consider least-squares regression with a linear GNN,* $f_W^C(A, X) = C(A)XW$ *parametrized by* $W$, *there always exists a condensed graph* $\mathcal{S}^*$, *such that the gradient matching objective for* $f^C$ *is achievable. However, if we train a new linear GNN* $f_W^{C'}(A, X)$ *with convolution matrix* $C'(A')$ *on* $\mathcal{S}^*$, *the relative error between the optimized model parameters of* $f_W^{C'}$ *on the real and condensed graphs is* $\|W_{C'}^{\mathcal{S}} - W_{C'}^{\mathcal{T}}\| / \|W_{C'}^{\mathcal{T}}\| \geq \max\{\sigma_{\max}(Q) - 1, 1 - \sigma_{\min}(Q)\}$, *where* $W_{C'}^{\mathcal{T}} = \arg\min_W \|\mathbf{y} - f_W^{C'}(A, X)\|_2^2$, $W_{C'}^{\mathcal{S}} = \arg\min_W \|\mathbf{y}' - f_W^{C'}(A', X')\|_2^2$, *and* $Q = \left(X^\top [C(A)]^\top [C(A)] X\right)\left(X^\top [C'(A)]^\top [C'(A)] X\right)^{-1}$.

## B    MORE PRELIMINARIES

In this section, we describe in greater details the types of data, downstream tasks, and neural network models that our *hyperparameter-calibrated dataset condensation* (HCDC) applies to. Moreover, we also religiously define the simplified *linear convolution regression problem* with least-square loss and linear convolution models, which is the assumed setup for Lemma 1 and Propositions 1 to 3.

### B.1    DOWNSTREAM TASKS

In Section 2 we have defined the downstream task that this paper mainly focus on, **node classification on graphs**. Where we are given a graph $\mathcal{T} = (A, X, \mathbf{y})$ with adjacency matrix $A \in \{0, 1\}^{n \times n}$, node features $X \in \mathbb{R}^{n \times d}$, node class labels $\mathbf{y} \in [K]^n$, and mutually disjoint node-splits $V_{\mathrm{train}} \bigcup V_{\mathrm{val}} \bigcup V_{\mathrm{test}} = [n]$, and the goal is to predict the node labels.

Here, to give a background on the *convolution neural networks* (CNNs) applications discussed in Section 2.3, we show as follows the settings above can be also used to describe **per-pixel classification on images** (e.g., for semantic segmentation) where CNNs are usually used.

For *per-pixel classification*, we are given a set of $\mathfrak{n}$ images of size $w \times h$, so the pixel values of the $j$-th image can be formatted as a tensor $\mathfrak{X}_j \in \mathbb{R}^{w \times h \times c}$ if there are $c$ channels. We are also given the pixel labels $\mathfrak{Y}_j \in [K]^{w \times h}$ for each image $j \in [\mathfrak{n}]$ and the mutually disjoint image-splits $I_{train} \bigcup I_{val} \bigcup I_{test} = [\mathfrak{n}]$. Clearly, we can reshape the pixel values and pixel labels of

the $j$-th image to $wh \times c$ and $wh$ respectively, and concatenate those matrices from all images. Following this, denoting $n = \mathfrak{n}wh$, we obtain the concatenated pixel value matrix $X \in \mathbb{R}^{n \times c}$ and the concatenated pixel label vector $\mathbf{y} \in [K]^n$. The image-splits are translated into pixel-level splits where $V_{train} = \{i \mid (j-1)wh \leq i \leq jwh, j \in I_{trian}\}$ (similar for $V_{val}$ and $V_{test}$) and $V_{train} \bigcup V_{val} \bigcup V_{test} = [n]$. We can also define the auxiliary adjacency matrix $A \in \{0,1\}^{n \times n}$ on the $n = \mathfrak{n}wh$ pixels, where $A$ is block diagonal $A = \text{diag}(A_1, \dots, A_{\mathfrak{n}})$ and $A_j \in \{0,1\}^{wh \times wh}$ is the assumed adjacency (e.g. a two-dimensional grid) of the $j$-th image.

## B.2 NEURAL NETWORK MODELS

This paper mainly focus on **graph neural networks** (GNNs) $f_{\theta,\lambda} : \mathbb{R}_{\geq 0}^{n \times n} \times \mathbb{R}^{n \times d} \to \mathbb{R}^{n \times K}$, where $\theta \in \Theta$ denotes the parameters and $\lambda \in \Lambda$ denotes the hyperparameters. In Section 2 we have seen that most GNNs can be interpreted as iterative convolution / message passing over nodes (Ding et al., 2021; Balcilar et al., 2021) where $X^{(0)} = X$ and $f(A, X) = X^{(L)}$, and for $l \in [L]$, the update-rule is,

$$X^{(l+1)} = \sigma\Big( C_{\alpha^{(l)}}(A) X^{(l)} W^{(l)} \Big), \tag{4}$$

where $C_{\alpha^{(l)}}(A)$ is the convolution matrix parametrized by $\alpha^{(l)}$, $W^{(l)}$ is the learnable linear weights, and $\sigma(\cdot)$ denotes the non-linearity. Thus the parameters $\theta$ consists of all $\alpha$'s (if they exist) and $W$'s, i.e., $\theta = [\alpha^{(0)}, \dots, \alpha^{(L-1)}, W^{(0)}, \dots, W^{(L-1)}]$.

More specifically, it is possible for GNNs to have more than one convolution filters per layer (Ding et al., 2021; Balcilar et al., 2021) and we may generalize Eq. (4) to,

$$X^{(l+1)} = \sigma\Big( \sum_{i=1}^{p} C_{\alpha^{(l,i)}}^{(i)}(A) X^{(l)} W^{(l,i)} \Big). \tag{5}$$

Within this common framework, GNNs differ from each other by the choice of convolution filters $\{C^{(i)}\}$, which can be either fixed or learnable. If $C^{(i)}$ is fixed, there is no parameters $\alpha^{(l,i)}$ for any $l \in [L]$. If $C^{(i)}$ is learnable, the convolution matrix relies on the learnable parameters $\alpha^{(l,i)}$ and can be different in each layers (thus should be denoted as $C^{(l,i)}$). Usually for GNNs, the convolution matrix depends on the parameters in two possible ways: (1) the convolution matrix $C^{(l,i)}$ is scaled by the scalar parameter $\alpha^{(l,i)} \in \mathbb{R}$, i.e., $C^{(l,i)} = \alpha^{(l,i)} \mathfrak{C}^{(i)}$ (e.g. GIN (Xu et al., 2018), ChebNet (Defferrard et al., 2016), and SIGN (Frasca et al., 2020)); or (2) the convolution matrix is constructed by node-level self-attentions $[C^{(l,i)}]_{ij} = h_{\alpha^{(l,i)}}\big(X_{i,:}^{(l)}, X_{j,:}^{(l)}\big)[\mathfrak{C}^{(i)}]_{i,j}$ (e.g., GAT (Veličković et al., 2018), Graph Transformers (Rong et al., 2020; Puny et al., 2020; Zhang et al., 2020)). Based on (Ding et al., 2021; Balcilar et al., 2021), we summarize the popular GNNs reformulated into the convolution over nodes / message-passing formula (Eq. (5)) in Table 4.

**Convolutional neural networks** can also be reformulated into the form of Eq. (5). For simplicity we only consider one-dimensional convolution neural network (1D-CNN) and the generalization to 2D/3D-CNNs is trivial. If we denote the constant cyclic permutation matrix (which corresponds to a unit shift) as $P \in \mathbb{R}^{n \times n}$, the update rule of a 1D-CNN with kernel size $(2K+1)$, $K \geq 0$ can be written as,

$$X^{(l+1)} = \sigma\Big( \sum_{k=-K}^{k=K} \alpha_k P^k X^{(l)} W^{(l,k)} \Big). \tag{6}$$

We will use this common convolution formula of GNNs (Eq. (5)) and 1D-CNNs (Eq. (6)) in Appendix B.4 and Proposition 1.

## B.3 OTHER TYPES OF DATA, TASKS, AND MODELS

In Appendices B.1 and B.2 we have discussed the formal definition of two possible tasks (1) node classification on graphs and (2) per-pixel classification on images, and reformulated many popular GNNs and CNNs into a general convolution form (Eqs. (5) and (6)). However, we want to note that the application of dataset condensation methods (including the standard dataset condensation (Wang et al., 2018; Zhao et al., 2020; Zhao & Bilen, 2021b) and our HCDC) is not limited by the specific types of data, tasks, and models.

| Model Name | Design Idea | Conv. Matrix Type | # of Conv. | Convolution Matrix |
|---|---|---|---|---|
| GCN[1] (Kipf & Welling, 2016) | Spatial Conv. | Fixed | 1 | $C = \widetilde{D}^{-1/2}\widetilde{A}\widetilde{D}^{-1/2}$ |
| SAGE-Mean[2] (Hamilton et al., 2017) | Message Passing | Fixed | 2 | $\begin{cases} C^{(1)} = I_n \\ C^{(2)} = D^{-1}A \end{cases}$ |
| GAT[3] (Veličković et al., 2018) | Self-Attention | Learnable | # of heads | $\begin{cases} \mathfrak{C}^{(s)} = A + I_n \text{ and} \\ h_{\boldsymbol{a}^{(l,s)}}^{(s)}(X_{i,:}^{(l)}, X_{j,:}^{(l)}) = \exp\left(\text{LeakyReLU}\left(\right.\right. \\ \left.\left. (X_{i,:}^{(l)}W^{(l,s)} \parallel X_{j,:}^{(l)}W^{(l,s)}) \cdot \boldsymbol{a}^{(l,s)}\right)\right) \end{cases}$ |
| GIN[1] (Xu et al., 2018) | WL-Test | Fixed + Learnable | 2 | $\begin{cases} C^{(1)} = A \\ \mathfrak{C}^{(2)} = I_n \text{ and } h_{\epsilon^{(l)}}^{(2)} = 1 + \epsilon^{(l)} \end{cases}$ |
| SGC[2] (Defferrard et al., 2016) | Spectral Conv. | Learnable | order of poly. | $\begin{cases} \mathfrak{C}^{(1)} = I_n, \mathfrak{C}^{(2)} = 2L/\lambda_{\max} - I_n, \\ \mathfrak{C}^{(s)} = 2\mathfrak{C}^{(2)}\mathfrak{C}^{(s-1)} - \mathfrak{C}^{(s-2)} \\ \text{and } h_{\theta^{(s)}}^{(s)} = \theta^{(s)} \end{cases}$ |
| ChebNet[2] (Defferrard et al., 2016) | Spectral Conv. | Learnable | order of poly. | $\begin{cases} \mathfrak{C}^{(1)} = I_n, \mathfrak{C}^{(2)} = 2L/\lambda_{\max} - I_n, \\ \mathfrak{C}^{(s)} = 2\mathfrak{C}^{(2)}\mathfrak{C}^{(s-1)} - \mathfrak{C}^{(s-2)} \\ \text{and } h_{\theta^{(s)}}^{(s)} = \theta^{(s)} \end{cases}$ |
| GDC[3] (Klicpera et al., 2019) | Diffusion | Fixed | 1 | $C = S$ |
| Graph Transformers[4] (Rong et al., 2020) | Self-Attention | Learnable | # of heads | $\begin{cases} \mathfrak{C}_{i,j}^{(s)} = 1 \text{ and } h_{(W_Q^{(l,s)}, W_K^{(l,s)})}^{(s)}(X_{i,:}^{(l)}, X_{j,:}^{(l)}) \\ = \exp\left(\frac{1}{\sqrt{d_{k,l}}}(X_{i,:}^{(l)}W_Q^{(l,s)})(X_{j,:}^{(l)}W_K^{(l,s)})^{\mathsf{T}}\right) \end{cases}$ |

[1] Where $\widetilde{A} = A + I_n$, $\widetilde{D} = D + I_n$. [2] $C^{(2)}$ represents mean aggregator. Weight matrix in (Hamilton et al., 2017) is $W^{(l)} = W^{(l,1)} \parallel W^{(l,2)}$. [3] Need row-wise normalization. $C_{i,j}^{(l,s)}$ is non-zero if and only if $A_{i,j} = 1$, thus GAT follows direct-neighbor aggregation. [4] The weight matrices of the two convolution supports are the same, $W^{(l,1)} = W^{(l,2)}$. [5] Where normalized Laplacian $L = I_n - D^{-1/2}AD^{-1/2}$ and $\lambda_{\max}$ is its largest eigenvalue, which can be approximated as 2 for a large graph. [6] Where $S$ is the diffusion matrix $S = \sum_{k=0}^{\infty} \theta_k \boldsymbol{T}^k$, for example, decaying weights $\theta_k = e^{-t}\frac{t^k}{k!}$ and transition matrix $\boldsymbol{T} = \widetilde{D}^{-1/2}\widetilde{A}\widetilde{D}^{-1/2}$. [7] Need row-wise normalization. Only describes the global self-attention layer, where $W_Q^{(l,s)}, W_Q^{(l,s)} \in \mathbb{R}^{f_l, d_{k,l}}$ are weight matrices which compute the queries and keys vectors. In contrast to GAT, all entries of $\mathfrak{C}_{i,j}^{(l,s)}$ are non-zero. Different design of Graph Transformers (Puny et al., 2020; Rong et al., 2020; Zhang et al., 2020) use graph adjacency information in different ways, and is not characterized here, see the original papers for details.

Table 4: Summary of GNNs formulated as generalized graph convolution.

For HCDC, we can follow the conventions in (Zhao et al., 2020) to define the train/validation losses on *iid* samples and define the notion of dataset condensation as learning a smaller synthetic dataset with less number of samples. Here we leave the readers to (Zhao et al., 2020) for formal definitions of condensation on datasets with *iid* samples.

More generally speaking, our HCDC can be applied as long as (1) the train and validation losses, i.e., $\mathcal{L}_{\mathcal{T}}^{train}(\theta, \lambda)$ and $\mathcal{L}_{\mathcal{T}}^{val}(\theta, \lambda)$ can be defined (as functions of the parameters and hyperparameters); and (2) we have an well-defined notion of the learnable synthetic dataset $\mathcal{S}$, (e.g., which includes prior-knowledge like what is the format of the synthetic data in $\mathcal{S}$ and how the same model $f_{\theta,\lambda}$ is applied).

## B.4 THE LINEAR CONVOLUTION REGRESSION PROBLEM

For the ease of theoretical analysis, in Lemma 1 and Propositions 1 to 3 we consider a simplified *linear convolution regression problem* as follows,

$$\theta^{\mathcal{T}} = \arg\min_{\theta=[\alpha,W]} \|C_\alpha(A)XW - \mathbf{y}\|^2 \tag{7}$$

where we are given continuous labels $\mathbf{y}$ and use sum-of-squares loss $\ell(\hat{y}, y) = \|\hat{y} - y\|_2^2$ instead of the cross entropy loss used for node/pixel classification. We also assume a linear GNN/CNN $f_{\theta=[\alpha,W]}(A, X) = C_\alpha(A)XW$ is used, where $C_\alpha(A)$ is the *convolution matrix* which depends on the adjacency matrix $A$ and the parameters $\alpha \in \mathbb{R}^p$, and $W$ is the learnable linear weights with $d$ elements (hence, the complete parameters consists of two parts, i.e., $\theta = [\alpha, W]$).

As explained in Appendix B.2, this *linear convolution* model $f_{\theta=[\alpha,W]}(A, X) = C_\alpha(A)XW$ already generalizes a wide variety of GNNs and CNNs. For example, it can represents the (single-layer) graph convolution network (GCN) (Kipf & Welling, 2016) whose convolution matrix is defined as $C(A) = \tilde{D}^{-\frac{1}{2}}\tilde{A}\tilde{D}^{-\frac{1}{2}}$ where $\tilde{A}$ and $\tilde{D}$ are the "self-loop-added" adjacency and degree matrix (for

GCNN there is no learnable parameters in $C_(A)$ so we omit $\alpha$). It also generalizes the one-dimensional *convolution neural network* (1D-CNN), where the convolution matrix is $C_\alpha(A) = \sum_{k=-K}^{k=K} [\theta]_k P^k$ and $P$ is the cyclic permutation matrix correspond to a unit shift.

It is important to note that although we considered this simplified *linear convolution regression problem* in some of our theoretical results, which is both convex and linear. We argue that most of the theoretical phenomena reflected by Lemma 1 and Propositions 1 to 3 can be generalized to the general non-convex losses and non-linear models; see Appendix C.4 for the corresponding discussions.

# C   PROOFS AND EXTENDED THEORETICAL RESULTS

In this section, we provide the proofs to the theoretical results Lemma 1 and Propositions 1 to 3 and Theorem 1, together with some extended theoretical discussions, including generalizing the *linear convolution regression problem* to non-convex losses and non-linear models (see Appendix C.4).

To proceed, please recall the *linear convolution regression problem* defined in Appendix B.4, the achievability of gradient-matching objective (Eq. (2)) defined as Assumption 1 in Section 2.3.

## C.1   VALIDITY OF STANDARD DATASET CONDENSATION

As the first step, we verify the validity of the standard dataset condensation (SDC) using the gradient-matching objective Eq. (2) for the *linear convolution regression problem*.

**Lemma 1.** *(Validity of SDC)  Consider least square regression with linear convolution model $f_W(A, X) = C(A)XW$ parameterized by $W$. If the gradient-matching objective of $f_W$ is achievable, then the optimizer on the condensed dataset $\mathcal{S}$, i.e., $W^{\mathcal{S}} = \arg\min_W \mathcal{L}_{\mathcal{S}}(W)$ is also optimal for the original dataset, i.e., $\mathcal{L}_{\mathcal{T}}(W^{\mathcal{S}}) = \min_W \mathcal{L}_{\mathcal{T}}(W)$.*

*Proof.* In the *linear convolution regression problem*, sum-of-squares loss is used, and $\mathcal{L}_{\mathcal{T}}(W) = \|CXW - \mathbf{y}\|_2^2$ (similarly $\mathcal{L}_{\mathcal{S}}(W) = \|C'X'W - \mathbf{y}'\|_2^2$ where $C' = C(A')$). We assume $X^\top C^\top CX \in \mathbb{R}^{d \times d}$ is invertible and we can apply the optimizer formula for ordinary least square (OLS) regression to find the optimizer $W^{\mathcal{T}}$ of $\mathcal{L}_{\mathcal{T}}(W)$ as,

$$W^{\mathcal{T}} = (X^\top C^\top CX)^{-1} X^\top C^\top \mathbf{y}.$$

Also, we can compute the gradients of $\mathcal{L}_{\mathcal{T}}(W)$ w.r.t $W$ as,

$$\nabla_W \mathcal{L}_{\mathcal{T}}(W) = 2X^\top C^\top (CXW - \mathbf{y}),$$

and similarly for $\nabla_W \mathcal{L}_{\mathcal{S}}(W)$.

Given the achievability of the gradient-matching objective of $f_W$, we know there exists a non-degenerate trajectory $(W_t^{\mathcal{S}})_{t=0}^{T-1}$ which spans the entire parameter space, i.e., $\text{span}(W_0^{\mathcal{S}}, \ldots, W_{T-1}^{\mathcal{S}}) = \mathbb{R}^d$, such that the gradient-matching loss (the objective of Eq. (2) without expectation) on this trajectory is 0. Assuming $D(\cdot, \cdot)$ is the $L_2$ norm (Zhao et al., 2020), this means,

$$\nabla_W \mathcal{L}_{\mathcal{T}}(W_t^{\mathcal{S}}) = \nabla_W \mathcal{L}_{\mathcal{S}}(W_t^{\mathcal{S}}) \quad \text{for} \quad t \in [T].$$

Substitute in the formula for the gradients $\nabla_W \mathcal{L}_{\mathcal{T}}(W)$ and $\nabla_W \mathcal{L}_{\mathcal{S}}(W)$, we then have,

$$X^\top C^\top (CXW_t^{\mathcal{S}} - \mathbf{y}) = X'^\top C'^\top (C'X'W_t^{\mathcal{S}} - \mathbf{y}') \quad \text{for} \quad t \in [T].$$

Since the set of $\{W_t^{\mathcal{S}}\}_{t=0}^{T-1}$ spans the complete parameter space $\mathbb{R}^d$, we can transform the set of vectors $\{\omega_t \cdot W_t^{\mathcal{S}}\}_{t=0}^{T-1}$ to the set of unit vectors $\{\mathbf{e}_i^d\}_{i=0}^{d-1} \in \mathbb{R}^d$ by a linear transformation. Meanwhile, the set of $T$ equations above can be transformed to,

$$X^\top C^\top (CX\mathbf{e}_i^d - \mathbf{y}) = X'^\top C'^\top (C'X'\mathbf{e}_i^d - \mathbf{y}') \quad \text{for} \quad i \in [d].$$

This directly leads to $X^\top C^\top CX = X'^\top C'^\top C'X'$ and $X^\top C^\top \mathbf{y} = X'^\top C'^\top \mathbf{y}'$.

Using the formula for the optimizers $W^{\mathcal{T}}$ and $W^{\mathcal{S}}$ above, we readily get,

$$W^{\mathcal{T}} = (X^\top C^\top CX)^{-1} X^\top C^\top \mathbf{y} = (X'^\top C'^\top C'X')^{-1} X'^\top C'^\top \mathbf{y}' = W^{\mathcal{S}}.$$

And hence,

$$\mathcal{L}_{\mathcal{T}}(W^{\mathcal{S}}) = \mathcal{L}_{\mathcal{T}}(W^{\mathcal{T}}) = \min_W \mathcal{L}_{\mathcal{T}}(W),$$

which concludes the proof. $\square$

Despite its simplicity, Lemma 1 directly verifies the validity of the gradient-matching formulation of standard dataset condensation on some specific learning problems. Although the gradient-matching formulation (Eq. (2)) is an efficient but weaker formulation than the bilevel formulation of SDC (Eq. (1)), we see it is strong enough for some the *linear convolution regression problem*.

## C.2  GENERALIZATION ISSUES OF SDC

Now, we move forward and focus on the generalization issues of (the gradient-matching formulation of) the standard dataset condensation (SDC) across GNNs.

To start with, we prove the successful generalization of SDC across 1D-CNNs as follows, which is very similar to the proof of Lemma 1.

*Proof of Proposition 1:* In Proposition 1, we consider one-dimensional linear convolution models $f^{2K+1}(X) = (\sum_{k=-K}^{k=K} \alpha^{(k)} P^k) X W$ parameterized by $\alpha \in \mathbb{R}^p$ and $W \in \mathbb{R}^d$ (where $p = 2K + 1$). If we denote,

$$C = \sum_{k=-K}^{k=K} \alpha^{(k)} P^k \quad \text{and} \quad \theta = [\alpha, W] \in \mathbb{R}^{p+d}$$

then from the proof of Lemma 1 we know the gradients of $\mathcal{L}_{\mathcal{T}}(W)$ w.r.t $W$ is again,

$$\nabla_W \mathcal{L}_{\mathcal{T}}(W) = 2X^\top C^\top (CXW - \mathbf{y}).$$

We know the achievability of the gradient-matching objective means there exists a non-degenerate trajectory $(\theta_t^{\mathcal{S}})_{t=0}^{T-1}$ which spans the entire parameter space, i.e., $\text{span}(\theta_0^{\mathcal{S}}, \ldots, \theta_{T-1}^{\mathcal{S}}) = \mathbb{R}^{p+d}$. By decomposing $\theta_t^{\mathcal{S}}$ into $[\alpha_t^{\mathcal{S}}, W_t^{\mathcal{S}}]$, we know that there exists $(\alpha_t^{\mathcal{S}})_{t=0}^{T-1}$ which spans $\mathbb{R}^p$ and there exists $(W_t^{\mathcal{S}})_{t=0}^{T-1}$ which spans $\mathbb{R}^d$.

Since the gradient-matching objective is minimized to 0 on $(W_t^{\mathcal{S}})_{t=0}^{T-1}$ which spans $\mathbb{R}^d$, following the same procedure as the proof of Lemma 1, we again obtain,

$$X^\top C^\top \mathbf{y} = X'^\top C'^\top \mathbf{y}'.$$

Mean while, since the same gradient-matching objective is also minimized to 0 on $(\alpha_t^{\mathcal{S}})_{t=0}^{T-1}$ which spans $\mathbb{R}^p$, we have,

$$X^\top \Big( \sum_{k=-K}^{k=K} (\alpha_t^{\mathcal{S}})^{(k)} P^k \Big)^\top \mathbf{y} = X'^\top \Big( \sum_{k=-K}^{k=K} (\alpha_t^{\mathcal{S}})^{(k)} P'^k \Big)^\top \mathbf{y}' \quad \text{for} \quad t \in [T].$$

Again by linear combining the above $T$ equations and because $(\alpha_t^{\mathcal{S}})_{t=0}^{T-1}$ can be transformed to the unit vectors in $\mathbb{R}^p$, we have,

$$X^\top (P^k)^\top \mathbf{y} = X'^\top (P'^k)^\top \mathbf{y}' \quad \text{for} \quad k = -K, \ldots, K.$$

Hence, for any new trajectory $(\alpha_t'^{\mathcal{S}})_{t=0}^{T-1}$ which spans $\mathbb{R}^{p'}$ where $p' = 2K' + 1$, by linear combining the above equations, we have,

$$X^\top \Big( \sum_{k=-K}^{k=K} (\alpha_t'^{\mathcal{S}})^{(k)} P^k \Big)^\top \mathbf{y} = X'^\top \Big( \sum_{k=-K}^{k=K} (\alpha_t'^{\mathcal{S}})^{(k)} P'^k \Big)^\top \mathbf{y}' \quad \text{for} \quad t \in [T'].$$

With similar procedure for the $X^\top C^\top CX$ part, we conclude that on the new trajectory $(\theta_t'^{\mathcal{S}})_{t=0}^{T-1}$

$$\nabla_W \mathcal{L}_{\mathcal{T}}(\alpha, W) = \nabla_W \mathcal{L}_{\mathcal{S}}(\alpha, W).$$

It remains to prove that on any new trajectory $\nabla_\alpha \mathcal{L}_{\mathcal{T}}(\alpha, W) = \nabla_\alpha \mathcal{L}_{\mathcal{S}}(\alpha, W)$. Only need to note that,

$$\nabla_{\alpha^{(k)}} \mathcal{L}_{\mathcal{T}}(\alpha, W) = 2W^\top X^\top P^k (CXW - \mathbf{y}).$$

Hence, by the $p$ equations above we can readily show,

$$X^\top P^k \mathbf{y} = X'^\top P'^k \mathbf{y}' \quad \text{for} \quad k = -K, \ldots, K.$$

Again with similar procedure for the $X^\top C^\top C X$ part, we finally can show that on the new trajectory $(\theta_t'^{\mathcal{S}})_{t=0}^{T-1}$

$$\nabla_\alpha \mathcal{L}_\mathcal{T}(\alpha, W) = \nabla_\alpha \mathcal{L}_\mathcal{S}(\alpha, W).$$

This concludes the proof. □

Then we focus on the linear GNNs, we want to verify the insight that the learned adjacency $A'$ of the condensed graph has "too many degrees of freedom" so that can easily overfit the gradient-matching objective, no matter what learned synthetic features $X'$ are. Again, the proof of Proposition 2 uses some results in the proof of Lemma 1.

*Proof of Proposition 2:* Now, we consider a linear GNN defined as $f(A, X) = C(A)XW$. From the proof of Lemma 1, we know that for the gradient-matching objective of $f$ to be achievable, it is equivalent to require that,

$$X^\top C^\top C X = X'^\top C'^\top C' X' \quad \text{and} \quad X^\top C^\top \mathbf{y} = X'^\top C'^\top \mathbf{y}',$$

where $C$ and $C'$ refer to $C(A)$ and $C(A')$ respectively.

Firstly we note that once we find $C'$ and $X'$ such that satisfy the first condition $X^\top C^\top C X = X'^\top C'^\top C' X'$, we can always find $\mathbf{y}' \in \mathbb{R}^c$ such that $X^\top C^\top \mathbf{y} = X'^\top C'^\top \mathbf{y}'$ since $X^\top C^\top \mathbf{y} \in \mathbb{R}$ is a scalar.

Now, we focus on finding the convolution matrix $C'$ and the node feature matrix $X'$ of the condensed synthetic graph to satisfy $X^\top C^\top C X = X'^\top C'^\top C' X'$. We assume $n \gg c \gg d$ and consider the diagonalization of $X^\top C^\top C X \in \mathbb{R}^{d \times d}$. Since $X^\top C^\top C X$ is positive semi-definite, it can be diagonalized as $X^\top C^\top C X = V S^2 V^\top$ where $V \in \mathbb{R}^d$ is an orthogonal matrix and $S \in \mathbb{R}^d$ is a diagonal matrix that $S = \text{diag}(s_1, \ldots, s_d)$.

For any (real) semi-unitary matrix $U \in \mathbb{R}^{c \times d}$ such that $U^\top U = I_d$, we can construct $C'X' = USV^\top \in \mathbb{R}^{c \times d}$ and we can easily verify they satisfy the condition,

$$X'^\top C'^\top C' X' = VSU^\top USV^\top = VS^2V^\top = X^\top C^\top C X.$$

Then since $X'$ is full ranked, for any $X'$, by considering the singular-value decomposition of $X'$, we see that we can always find a convolution matrix $C'$ such that $C'X' = USV$ and this concludes the proof. □

Finally, we use some results of Proposition 2 to prove Proposition 3, the failure of SDC when generalizating across GNNs.

*Proof of Proposition 3:* We prove by two steps.

For the first step, we aim to show that there always exist a condensed synthetic dataset $\mathcal{S}$ such that achieves the gradient-matching objective but the learned adjacency matrix $A' = I_c$ is the identity matrix. Clearly this directly follows form the proof of Proposition 2, where we only require $C'X' = USV$ (see the proof of Proposition 2 for details). If the learned adjacency matrix $A' = I_c$, the for any GNNs, the corresponding convolution matrix $C'$ is also (or proportional to) identity, thus we only need to set the learned node feature matrix $X' = USV$ to satisfy the condition. The first step is proved.

For the second step, we evaluate the relative estimation error of the optimal parameter when transfer to a new GNN $f_W^{\mathcal{C}}$ with convolution filter $\mathcal{C}(\cdot)$, i.e., $\|W_{\mathcal{C}}^{\mathcal{S}} - W_{\mathcal{C}}^{\mathcal{T}}\| / \|W_{\mathcal{C}}^{\mathcal{T}}\|$. Using the formula for the optimal parameter in the proof of Lemma 1 again, we have,

$$W_{\mathcal{C}}^{\mathcal{T}} = (X^\top \mathcal{C}^\top \mathcal{C} X)^{-1} X^\top \mathcal{C}^\top \mathbf{y},$$

and

$$W_{\mathcal{C}}^{\mathcal{S}} = (X'^\top \mathcal{C}'^\top \mathcal{C}' X)^{-1} X'^\top \mathcal{C}'^\top \mathbf{y}',$$

where $\mathcal{C}' = \mathcal{C}(A') = \mathcal{C}(I_c) = C(I_c)$ (the last equation use the fact that the convolution matrix of GNNs are the same if the underlying graph is identity).

Moreover, by the validity of SDC on $f_W^C$, we know, (see the proof of Lemma 1 for details),

$$X'^\top C'^\top C' X' = X^\top C^\top C X \quad \text{and} \quad X'^\top C'^\top \mathbf{y}' = X^\top C^\top \mathbf{y}$$

Thus, altogether we derive that $X'^\top \mathcal{C}'^\top \mathcal{C}' X = X^\top C^\top C X$ and $X'^\top \mathcal{C}'^\top \mathbf{y}' = X^\top C^\top \mathbf{y}$. And therefore,

$$W_{\mathcal{C}}^{\mathcal{S}} = (X^\top C^\top C X)^{-1} X^\top C^\top \mathbf{y}.$$

Now, note that,

$$\|W_{\mathcal{C}}^{\mathcal{S}} - W_{\mathcal{C}}^{\mathcal{T}}\|/\|W_{\mathcal{C}}^{\mathcal{T}}\|$$

$$= \left\|\left(\left(X^\top [C(A)]^\top [C(A)]X\right)\left(X^\top [\mathcal{C}(A)]^\top [\mathcal{C}(A)]X\right)^{-1}\right) - I_d\right)X^\top C^\top \mathbf{y}\right\|/\|X^\top C^\top \mathbf{y}\|$$

$$\geq \max\{\sigma_{\max}(Q) - 1, 1 - \sigma_{\min}(Q)\}$$

where $Q = \left(X^\top [C(A)]^\top [C(A)]X\right)\left(X^\top [\mathcal{C}(A)]^\top [\mathcal{C}(A)]X\right)^{-1}$. This concludes the proof. $\square$

## C.3 Validity of HCDC

Finally, we complete the proof of Theorem 1 with more detials.

*Proof of Theorem 1:* Firstly, we prove the *necessity* by contradiction.

If there exists $\lambda_0 \in \tilde{\Lambda}$ s.t. the two gradient vectors are not aligned at $\lambda_0$, then there exists small perturbation $\Delta\lambda_0$ such that $\mathcal{L}_{\mathcal{T}}^*(\lambda_0 + \Delta\lambda_0) - \mathcal{L}_{\mathcal{T}}^*(\lambda_0)$ and $\mathcal{L}_{\mathcal{S}}^*(\lambda_0 + \Delta\lambda_0) - \mathcal{L}_{\mathcal{S}}^*(\lambda_0)$ have different signs.

Secondly, we prove the *sufficiency* by path-integration.

For any pair $\lambda_1 \neq \lambda_2 \in \tilde{\Lambda}$, we have a path $\gamma(\lambda_1, \lambda_2) \in \tilde{\Lambda}$ from $\lambda_2$ and $\lambda_1$, then integrating hypergradients $\nabla_\lambda \mathcal{L}_{\mathcal{T}}^*(\lambda)$ along the path recovers the hyperparameter-calibration condition. More specifically, along the path we have $\mathcal{L}_{\mathcal{T}}^*(\lambda_1) - \mathcal{L}_{\mathcal{T}}^*(\lambda_2) = \int_{\gamma(\lambda_1, \lambda_2)} \nabla_\lambda \mathcal{L}_{\mathcal{T}}^*(\lambda)d\lambda$ (similar for $\nabla_\lambda \mathcal{L}_{\mathcal{S}}^*(\lambda)$). Thus we have,

$$(\mathcal{L}_{\mathcal{T}}^*(\lambda_1) - \mathcal{L}_{\mathcal{T}}^*(\lambda_2))(\mathcal{L}_{\mathcal{S}}^*(\lambda_1) - \mathcal{L}_{\mathcal{S}}^*(\lambda_2))$$

$$= \left(\int_{\gamma(\lambda_1, \lambda_2)} \nabla_\lambda \mathcal{L}_{\mathcal{T}}^*(\lambda)d\lambda\right)\left(\int_{\gamma(\lambda_1, \lambda_2)} \nabla_\lambda \mathcal{L}_{\mathcal{S}}^*(\lambda)d\lambda\right)$$

$$\geq \int_{\gamma(\lambda_1, \lambda_2)} \left\langle\sqrt{\nabla_\lambda \mathcal{L}_{\mathcal{T}}^*(\lambda)}, \sqrt{\nabla_\lambda \mathcal{L}_{\mathcal{S}}^*(\lambda)}\right\rangle d\lambda$$

$$\geq 0$$

the second last inequality by Cauchy-Schwarz inquality and the last inequality by $\cos(\nabla_\lambda \mathcal{L}_{\mathcal{T}}^*(\lambda), \nabla_\lambda \mathcal{L}_{\mathcal{T}}^*(\lambda)) = 0$ for any $\lambda \in \gamma(\lambda_1, \lambda_2) \in \tilde{\Lambda}$.

This concludes the proof. $\square$

## C.4 Generalize to Non-Convex and Non-Linear Case

Although the results above are obtained for least squares loss and linear convolution model, it *still reflects the nature of general non-convex losses and non-linear models*. Since dataset condensation is effectively matching the local minima $\{\theta^{\mathcal{T}}\}$ of the original loss $\mathcal{L}_{\mathcal{T}}^{train}(\theta, \psi)$ with the local minima $\{\theta^{\mathcal{S}}\}$ of the condensed loss $\mathcal{L}_{\mathcal{S}}^{train}(\theta, \psi)$, within the small neighborhoods surrounding the pair of local minima $(\theta^{\mathcal{T}}, \theta^{\mathcal{S}})$, we can approximate the non-convex loss and non-linear model with a convex/linear one respectively. Hence the generalizability issues with convex loss and liner model may hold.

# D Extended Related Work

This section contains the extensive discussions of many related work/areas which cannot be fitted into the main paper due to the page limit.

## D.1 Dataset Condensation and Coreset Selection

Firstly, we review the two main approaches to reducing the training set size while preserving model performance.

**Dataset condensation** (or distillation) is first proposed in (Wang et al., 2018) as a learning-to-learn problem by formulating the network parameters as a function of synthetic data and learning them through the network parameters to minimize the training loss over the original data. However, the nested-loop optimization precludes it scaling up to large-scale in-the-wild datasets. Zhao et al. (2020) alleviate this issue by enforcing the gradients of the synthetic samples w.r.t. the network weights to approach those of the original data, which successfully alleviates the expensive unrolling of the computational graph. Based on the meta-learning formulation in (Wang et al., 2018), Bohdal et al. (2020) and Nguyen et al. (2020; 2021) propose to simplify the inner-loop optimization of a classification model by training with ridge regression which has a closed-form solution, while Such et al. (2020) model the synthetic data using a generative network. To improve the data efficiency of synthetic samples in gradient-matching algorithm, Zhao & Bilen (2021a) apply differentiable Siamese augmentation, and Kim et al. (2022) introduce efficient synthetic-data parametrization. Recently, a new distribution-matching framework (Zhao & Bilen, 2021b) proposes to match the hidden features rather than the gradients for fast optimization, but may suffer from performance degradation compared to gradient-matching (Zhao & Bilen, 2021b), where Kim et al. (2022) provide some interpretation.

**Graph condensation** (Jin et al., 2021) achieves the state-of-the-art performance for preserving GNNs' performance on the simplified graph. However, Jin et al. (2021) only adapt the gradient-matching algorithm of dataset condensation Zhao et al. (2020) to graph data, together with a MLP-based generative model for edges (Anand & Huang, 2018; Simonovsky & Komodakis, 2018), leaving out several major issues on efficiency, performance, and generalizability. Subsequent work aims to apply the more efficient distribution-matching algorithm (Zhao & Bilen, 2021b; Wang et al., 2022) of dataset condensation to graph (Liu et al., 2022) or speed up gradient-matching graph condensation by reducing the number of gradient-matching-steps (Jin et al., 2022). While the efficiency issue of graph condensation is mitigated (Jin et al., 2022), the performance degradation on medium- and large-sized graphs still renders graph condensation practically meaningless. Our HCDC is specifically designed for repeated training in architecture search, which is, in contrast, well-motivated.

**Coreset selection** methods choose samples that are important for training based on heuristic criteria, for example, minimizing the distance between coreset and whole-dataset centers (Chen et al., 2010; Rebuffi et al., 2017), maximizing the diversity of selected samples in the gradient space (Aljundi et al., 2019), discovering cluster centers (Sener & Savarese, 2018), and choosing samples with the largest negative implicit gradient (Borsos et al., 2020). *Forgetting* (Toneva et al., 2018) measures the forgetfulness of trained samples and drops those that are not easy to forget. *GraNd* (Paul et al., 2021) selects the training samples that contribute most to the training loss in the first few epochs. *Prism* (Kothawade et al., 2022) select samples to maximize submodular set-functions which are combinatorial generalizations of entropy measures (Iyer et al., 2021). Recent benchmark (Guo et al., 2022) of a variety of coreset selection methods for image classification indicates *Forgetting*, *GraNd*, and *Prism* are among the best performing corset methods but still evidently underperform the dataset condensation baselines. Although coreset selection can be very efficient, most of the methods above suffer from three major limitations: (1) their performance is upper-bounded by the information in the selected samples; (2) most of them do not directly optimize the synthetic samples to preserve the model performance; and (3) most of them select samples incrementally and greedily, which are short-sighted.

## D.2 GRAPH REDUCTION

Secondly, we summarize the traditional graph reduction method for graph neural network training.

**Graph coreset selection** is a non-trivial generalization of the above method coreset methods given the non-*iid* nature of graph nodes and the non-linearity nature of GNNs. The very few off-the-shelf graph coreset algorithms are designed for graph clustering (Baker et al., 2020; Braverman et al., 2021) and are not optimal for the training of GNNs.

**Graph sampling** methods (Chiang et al., 2019; Zeng et al., 2019) can be as simple as uniformly sampling a set of nodes and finding their induced subgraph, which is understood as a graph-counterpart of uniform sampling of *iid* samples. However, most of the present graph sampling algorithms (e.g., ClusterGCN (Chiang et al., 2019) and GraphSAINT (Zeng et al., 2019)) are designed for sampling multiple subgraphs (mini-batches), which forms a cover of the original graph for training GNNs

with memory constraint. Therefore those graph mini-batch sampling algorithms are effectively graph partitioning algorithms and not optimized to find just one representative subgraph.

**Graph sparsification** (Batson et al., 2013; Satuluri et al., 2011) and **graph coarsening** (Loukas & Vandergheynst, 2018; Loukas, 2019; Huang et al., 2021; Cai et al., 2020) algorithms are usually designed to preserve specific graph properties like graph spectrum and graph clustering. Such objectives are often not aligned with the optimization of downstream GNNs and are shown to be sub-optimal in preserving the information to train GNNs well (Jin et al., 2021).

### D.3  OTHER RELATED AREAS

Lastly, we list two important relevant areas to this work, implicit differentiation methods based on the implicit function theorem (IFT), and the differentiable neural architecture search (NAS) algorithms.

**Implicit differentiation** methods apply the implicit function theorem (IFT) to the nested-optimization problems (Ochs et al., 2015; Wang et al., 2019). The IFT requires inverting the training Hessian with respect to the network weights, where early work either computes the inverse explicitly (Bengio, 2000; Larsen et al., 1996) or approximates it as the identity matrix (Luketina et al., 2016). Conjugate gradient (CG) is applied to invert the Hessian approximately (Pedregosa, 2016), but is difficult to scale to deep networks. Several methods have been proposed to efficiently approximate Hessian inverse, for example, 1-step unrolled differentiation (Luketina et al., 2016), Fisher information matrix (Larsen et al., 1996), NN-structure aided Kronecker-factored inversion (Martens & Grosse, 2015). Lorraine et al. (2020) use the Neumann inverse approximation, which is a stable alternative to CG (Shaban et al., 2019) and successfully scale gradient-based bilevel-optimization to large networks with constant memory constraint. It is shown that unrolling differentiation around locally optimal weights for $i$ steps is equivalent to approximating the Neumann series inverse approximation up to the first $i$ terms.

**Differentiable NAS** methods, e.g., DARTS (Liu et al., 2018) explore the possibility of transforming the discrete neural architecture space into a continuously differentiable form and further uses gradient optimization to search the neural architecture. DARTS follows a cell-based search space (Zoph et al., 2018) and continuously relaxes the original discrete search strategy. Despite its simplicity, several work cast double on the effectiveness of DARTS (Li & Talwalkar, 2020; Zela et al., 2019). SNAS (Xie et al., 2018) points out that DARTS suffers from the unbounded bias issue towards its objective, and it remodels the NAS and leverages the Gumbel-softmax trick (Jang et al., 2017; Maddison et al., 2017) to learn the exact architecture parameter. Differentiable NAS techniques have also been applied to graphs to automatically design data-specific GNN architectures (Wang et al., 2021; Huan et al., 2021).

## E  IMPLEMENTATION DETAILS

In this section we list more implementation details on the experiments in Section 6.

For the synthetic experiments on CIFAR-10, we randomly split the CIFAR-10 images into $M = 20$ splits and perform cross validation. For the baseline methods (Random, SDC-GM, SDC-DM), the dataset condensation is performed independently for each split. For HCDC, we first condense the training set of the synthetic dataset by SDC-GM. Then, we learn a separate validation set of with $1/M$-size of the training set and train with the HCDC objective on the $M$-HPO trajectories as described in Section 4. We report the correlation between the ranking of splits (in terms of their validation performance on this split). For the Early-Stopping method, we only train the same number iterations as the other methods (with the same batchsize), which means there are only $c/n * 500$ epochs.

For the experiments about finding the best convolution filter on (large) graphs, we create the set of ten candidate convolution filters as (see Table 4 for definitions and references) GCN, SAGE-Mean, SAGE-Max, GAT, GIN-$\epsilon$, GIN-0, SGC(K=2), SGC(K=3), ChebNet(K=2), ChebNet(K-3). The implementations are provided by PyTorch Geometric https://pytorch-geometric.readthedocs.io/en/latest/modules/nn.html. We also select the GNN width from $\{128, 256\}$ and the GNN depth from $\{2, 4\}$ so there are $10 \times 2 \times 2 = 40$ models in total.

For the experiments about speeding up off-the-shelf graph architecture search algorithms, we adopt GraphNAS (Gao et al., 2019) togher with their proposed search space from their official repository `https://github.com/GraphNAS/GraphNAS`. We apply to Ogbn-arxiv with condensation ratio $c_{train}/n = 0.5\%$.

