# OpenReview forum: "Faster Hyperparameter Search for GNNs via Calibrated Dataset Condensation"
_ICLR.cc/2023/Conference — Submitted to ICLR 2023_

### Official Review · Reviewer_RKbd · 2022-10-22

**Confidence:** 4
**Correctness:** 3
**Technical Novelty And Significance:** 2
**Empirical Novelty And Significance:** 3
**Recommendation:** 3

**Clarity, Quality, Novelty And Reproducibility:**

The overall quality of the paper is good. They conduct theoretical analysis and provide motivation for their work. However, it is not easy to follow the paper due to the complex and inconsistent notations. Also, the algorithmic details are absent and it is hard to fully understand their algorithm in detail. I believe applying hypergradient to dataset condensation is novel.

**Strength And Weaknesses:**

**Strength**
- The proposed idea matching the hyperparameter gradient is novel.
- The paper presents the problems of the existing methods theoretically and experimentally.
- The experimental results on GNN convolutional filter (Table 3) are promising.

**Weakness**
- **It is hard to follow the paper.** There are too many notations that are often not consistent. For example, (page 3) $\theta^S(S,\theta_0)$ and $\theta^T(\theta_0)$, (page 5) $\lambda^S$ and $\lambda^S(S)$. Why do the same notations have a different number of arguments? This makes me confused.
Also, the paper considers various hyperparameters (network type, width, learning rate; on page 5), but their mathematical notations of the paper only cover a very limited class of hyperparameters: $f_{\theta, \lambda}:R^{n\times n}\times R^{n\times d} \rightarrow R^{n\times K}$, where $\lambda$ is a hyperparameter. Based on this, they define the loss function as $L(\theta, \lambda)$, but general hyperparameters cannot be the argument of the loss function. For example, the learning rate is the argument of the learning algorithm, not the loss function. This inconsistency between the claims and explanations reduces the readability of the paper.
- **Algorithm is not described in detail.**  The paper uses the existing hypergradient method but differentiating the hypergradient is not trivial (see eq. (IFT) on page 7). However, the authors do not elaborate on the process. It would be informative if the authors provide the complexity analysis of the algorithm and error analysis by hypergradient approximation.
Also, the authors should summarize and present the final algorithm, which is currently absent. It is unclear how the synthetic training set and validation set are optimized. (Do they optimized sequentially?)
- **Experimental analyses are insufficient.** They conduct a synthetic experiment, finding the best cross-validation fold, but it is hard to understand the meaning of this experiment. (HCDC optimizes the validation set, and it is unclear what is the evaluation metric). Rather than synthetic experiments, the experiments in realistic settings (e.g., other types of hyperparameters and other domains or tasks) will be much more informative.
Also, there is no computational cost analysis of the condensation algorithm. Figure 4 compares the search process speed on synthetic and original data, but it is unfair because it does not consider the time for optimizing the synthetic set.

Further questions.
- The proposed method requires calculating hypergradient on the original training set. What is the benefit of the proposed method over the method directly using the hypergradient to optimize the hyperparameter?




**Summary Of The Paper:**

This paper proposes the graph condensation method for hyperparameter search. The proposed HCDC algorithm optimizes synthetic validation data as well as training data so that the obtained optimal hyperparameter on synthetic data shows good performance on the original validation set. They theoretically analyze previous approaches and verify existing methods can overfit to specific models and fail to generalize over other models. To overcome this issue, the paper proposes a new objective that matches hyperparameter gradients. The paper shows the effectiveness of their method on GNN convolutional filter search.

**Summary Of The Review:**

The paper proposes a novel approach on dataset condensation by matching hypergradient. There are some promising experimental results. However, the paper requires consistent and more concise notations for readability. Also, the final algorithm is not described in detail and the paper lacks some important experimental analysis, e.g., time complexity, optimization results of other hyperparameters, comparison to hyperparameter optimization by hypergradient. Due to these reasons, it is hard to understand their method fully.

-------

**Post rebuttal**

After reading the authors rebuttals, I still have some major concerns.
- In response 4, authors said their main contribution is to find a condensed dataset that preserves the validation performance ranking of architectures, and thus it is not directly comparable to hypergradient optimization methods. However, I still believe the paper should compare with the hypergradient optimization methods to demonstrate the effectiveness of the proposed approach. (the proposed method requires to compute the hypergradient with the original training set, so if the direct use of the hypergradient is more effective, then why should we use the proposed approach for NAS?
- In response 3.2, authors said it takes about 531 seconds for condensation before NAS. However, this is much larger than the time spent for search in Figure 4 (140 seconds) and it is unclear whether the proposed method is effective. Authors should reflect this on Figure 4.
- In response 1.2, authors mention that the focus of this paper is on the architecture hyperparameters, not general hyperparameters (learning rate). Then, the terms like HPO in the paper seems not appropriate, and I think the writing should be revised by only considering NAS, not HPO.

From the reasons above, I maintain my score.

---

### Official Review · Reviewer_K3vn · 2022-10-25

**Confidence:** 4
**Correctness:** 3
**Technical Novelty And Significance:** 3
**Empirical Novelty And Significance:** 3
**Recommendation:** 5

**Clarity, Quality, Novelty And Reproducibility:**

The paper is well written and well structured. The problem
is well explained, the proposed method is plausible, and
experiments look promising.

Smaller points:
1. I did not understand how the graph condensation works from reading
  just the paper and had to resort to Jin et al. 2021. Maybe explaining
  the training process at least in the appendix would make the paper more
  self contained.

2. Several typos should be fixed, e.g.,
  - p.9 "methods which does"
  - p.9 "we random split"


**Strength And Weaknesses:**

s1. interesting problem setting: graph condensation.
s2. clearly identified limitation in current work (overfitting hyperparameters)
  and plausible approach to overcome it.
s3. good experimental results.

w1. differences in the experiments w.r.t. published work.
w2. claim of training speedup not substantiated.
w3. the extended hyperparameter search space \tilde\Lambda is not clear
  to me.


**Summary Of The Paper:**

The paper addresses the problem of node classification in graphs,
esp. via graph condensation, i.e., finding a smaller graph on which
models can be learnt and perform well on the whole graph
nevertheless. The authors show that previous approaches (Jin et al.
2021) that match loss gradients do not work well across different
graph neural network (GNN) architectures. Therefore they
propose to match validation loss hyperparameter gradients
instead. They apply a method from the literature (Lorrainne et al. 2020)
to compute those gradients efficiently. In experiments they
show that their method better ranks hyperparameters w.r.t.
their performance on the whole graph and outperforms
existing baselines on four different datasets by up to 1.9
(absolute) %.


**Summary Of The Review:**

I see three issues:

w1. differences in the experiments w.r.t. published work.
  In the experiments, esp. tab. 3, performances are compared
  against the baseline model GCond (Jin et al. 2021), but there are
  major differences to the published numbers in the baseline paper, e.g.
  a. the dataset Flickr is dropped. Often different reduction ratios
      are used.
  b. the whole graph performance is different, e.g., for
      ogbn-arxiv you have 73.2, they have 71.4.
  c. the performance for the random baseline and for the baseline
      GCond differ, e.g., for ogbn-arix 0.25% and 0.5% you report
      70.5 and 71.1, they report 63.2 and 64.0.

  Is there an experimental condition changed that explains these
  differences? And if so, could you reproduce their results and
  compare head-to-head also?

w2. claim of training speedup not substantiated.

  The authors argue that by dataset condensation they could speed
  up training.
  While no one doubts that training on smaller graphs will take less time
  than training on larger graphs, to substantiate the claim that dataset
  condensation speeds up the training process one would have to compare
  the sum of the training time on the smaller graph and the time needed
  for its condensation to the training time on the larger graph. Is there
  really a speedup? And if so, how large is it?

w3. The extended hyperparameter search space \tilde\Lambda is not clear
  to me.

  What exactly is it, as a space, \R^p ? And how do you map
  between its elements and the p many hyperparameters in a
  differentiable manner?

---

### Official Review · Reviewer_ny5k · 2022-10-27

**Confidence:** 4
**Correctness:** 2
**Technical Novelty And Significance:** 3
**Empirical Novelty And Significance:** 3
**Recommendation:** 5

**Clarity, Quality, Novelty And Reproducibility:**

## Questions

* Why did you restrict a message passing algorithm in a graph neural network as a convolution operation? Is it necessary for developing your algorithm?

* How did you implement your optimization strategy in practice?

* For Proposition 3, does this statement support failed generalization across graph neural networks appropriately? If a graph becomes bigger, is not it naturally getting bigger? It implies that it only says the size of graph, instead of the lower bound of errors.

**Strength And Weaknesses:**

Here the strengths and weaknesses of this work are described.

## Strengths

It solves an interesting problem regarding graph neural networks in the perspective of hyperparameter optimization.

The arguments of this work is well-supported by thorough analyses and diverse evidences.

## Weaknesses

I think that presentation and writing can be improved more. The current version is okay, but some expressions need to be polished. In particular, there are too many terms with a dash "-", which makes sentences complicated. Also, a calligraphic "C", which stands for another convolution filter, is somewhat confusing with "C". I know that you can prefer your own expressions to write your paper, but these can be improved regardless of my suggestions. Moreover, I think "test", "cos", or any similar terms should be a regular font; please check them out carefully.

**Summary Of The Paper:**

This paper solves a problem of hyperparameter search for a graph neural network using calibrated dataset condensation. The authors propose a novel hyperparameter-calibrated dataset condensation framework by matching hyperparameter gradients in synthetic validation data generation. Unlike the standard dataset condensation, the proposed method is more robust for an overfitting issue, in particular for graph structures. Finally, the experimental results are demonstrated to show the validity of the proposed method.

**Summary Of The Review:**

Please check the above text boxes.

---

### Official Review · Reviewer_C3Kq · 2022-10-30

**Confidence:** 3
**Correctness:** 3
**Technical Novelty And Significance:** 3
**Empirical Novelty And Significance:** 2
**Recommendation:** 3

**Clarity, Quality, Novelty And Reproducibility:**

The paper is hard to read due to large amount of equations without much explanations. Also the description of the experiment setting seems confusing.

The proposed method seems novel.



**Strength And Weaknesses:**

Strength:

The task is novel and there are many potential applications for the use cases of the proposed method.

Weakness:

The paper is hard to read and have many confusing parts:

1) The paper is about hyperparameter search on GNN, but the experiment is on Cifar10.

2) The proposed method seems be used for constructing validation data only. According to the paper, the training data condensation is using other graph condensation methods. This brings the question of why not using the proposed method for training graph condensation? and what if performing NAS on the distilled graph generated by other graph condensation methods to search for hyperparameters?

3) The experiment result also seems confusing. For example, in Table 3, are the results on the distilled training graph as the ratio is on the n_train? to my understanding, the training graph is generated by other graph condensation method. Then how to choose the graph condensation method for training?






**Summary Of The Paper:**

This paper proposed a new graph condensation method (HCDC) for fast hyperparameter/architecture search by aligning the hyperparameter gradients. As in the paper, the proposed method is theoretically and experimentally proven to be effective on preserving the validation performance rankings of GNNs.

**Summary Of The Review:**

I think the task is interesting, and due to the readability of the paper, it is hard to justify the contribution.

---

### Decision · Program_Chairs · 2023-01-20

**Decision:**

Reject

**Justification For Why Not Higher Score:**

* Substantial reorganisation of the paper needed to better focus its narrative. Right now, too much material tends to be touched on, which leads to confusion and lack of clarity.
* Additional baselines required in the experiments to demonstrate the claimed efficiency

**Justification For Why Not Lower Score:**

N/A

**Metareview: Summary, Strengths And Weaknesses:**

The reviewers and meta reviewer all carefully checked and discussed the rebuttal. They thank the authors for their response and their efforts during the rebuttal phase.
The response helped resolve some concerns (e.g., discussions of the split values compared with Jin et al., 2021).

The reviewers and meta reviewer all acknowledge that the submission tackles an interesting problem (e.g., _“interesting problem setting: graph condensation”_) with some novel component (e.g., _“The proposed idea matching the hyperparameter gradient is novel”_).

At this stage, however, there are still some arguably important aspects in the submission that would benefit from further consolidations/investigations, for instance:

(i) The biggest current concern of the submission lies in the fact that it tries to cover too much material, which leads to confusion and a lack of clarity. For instance, in Section 2.3, the readers need to unpack the rich content of an entire theoretical analysis that is essentially in the appendix. Similarly, there are several features of the approach that are only briefly discussed while they would deserve more insights (e.g., the “supernet” is very little discussed while it is marked a key factor in the conclusion Section 7 “...how the supernet generalize to unseen architectures…“). Another case in point is the design of the “extended search space” which is far from a trivial detail and would thus benefit from further investigations.
As a suggestion, the paper could be sliced/reorganised to extract a more focused narrative. Perhaps, centering the paper around the idea of the matching of the hyperparameter gradient in dataset condensation would be sufficient, without having to speak about GNNs (e.g., looking at standard neural networks with common hyperparameters like L2 regularization, etc.). A follow-up paper could then be envisioned with the GNN part and the associated theory, this time centrally exposed.

(ii) In the current submission, adding other simple baselines such as multi-fidelity HPO approaches (BOHB, hyperband) would help demonstrate the claimed efficiency. Moreover, it should further be discussed/studied the fact that random selection is very close (within one standard error apparently) to the proposed approach on several datasets (not in terms of rank preservation, but final performance). This is happening despite the fact that the random selection is substantially simpler.

(iii) In spite of the provided replies, several design choices in the submission remain convoluted. For instance, it is difficult to understand why we need both HCDC and SDC. Moreover, the discussion about the assessment of the condensation overhead is also still unclear.

Because of the extremely competitive landscape of the submissions this year, the paper remains under the cut and has ultimately not been selected for acceptance.
We are convinced that the suggestions above will help strengthen the paper for a future resubmission, which the reviewers and meta reviewer all encourage.